# Ocean carbon cycling during the past 130,000 years - a pilot study on inverse paleoclimate record modelling

Christoph Heinze[1,2,3], Babette A.A. Hoogakker[4], and Arne Winguth[5]

[1]Geophysical Institute, University of Bergen, Allégaten 70, 5007 Bergen, Norway
[2]Uni Climate, Uni Research, Allégaten 55, 5007 Bergen, Norway
[3]Bjerknes Centre for Climate Research, Bergen, Norway
[4]Department of Earth Sciences, University of Oxford, South Parks Road, Oxford OX1 3AN, UK
[5]Department of Earth and Environmental Sciences, University of Texas Arlington, P.O. Box 19049, Arlington, TX 76019, USA

*Correspondence to:* Christoph Heinze (christoph.heinze@uib.no)

**Abstract.** What role did changes in marine carbon cycle processes and calcareous organisms play in glacial-interglacial variation in atmospheric $pCO_2$? In order to answer this question, we explore results from an ocean biogeochemical general circulation model. We attempt to systematically reconcile model results with time dependent sediment core data from the observations. For this purpose, we fit simulated sensitivities of oceanic tracer concentrations to changes in governing carbon cycle parameters to measured sediment core data. We assume that the time variation of the governing carbon cycle parameters follows the general pattern of the glacial-interglacial deuterium anomaly. Our analysis provides an independent estimate of a maximum mean sea surface temperature drawdown of about 5°C and a maximum outgassing of the land biosphere by about 430 PgC at the last glacial maximum as compared to preindustrial times. The overall fit of modelled paleoclimate tracers to observations, however, remains quite weak, indicating the potential of more detailed modelling studies to fully exploit the information stored in the paleo-climatic archive. This study confirms the hypothesis that a decline in ocean temperature and a more efficient biological carbon pump in combination with changes in ocean circulation are the key factors for explaining the glacial $CO_2$ drawdown. The analysis suggests that potential changes in the export rain ratio $POC:CaCO_3$ may not have a substantial imprint on the paleo-climatic archive. The use of the last glacial as an inverted analogue to potential ocean acidification impacts thus may be quite limited. A strong decrease in $CaCO_3$ export production could potentially contribute to the glacial $CO_2$ decline in the atmosphere, but remains hypothetical.

## 1 Introduction

The drawdown of the atmospheric $CO_2$ mixing ratio from 280-290 ppm to 180 ppm has become well known since the mid 1980s, following the analysis of the Antarctic ice cores (Neftel et al. (1982), Barnola et al. (1987), Siegenthaler et al. (2005)). Despite a series of attempts to explain this major fluctuation in the key atmospheric greenhouse gas $CO_2$, an overall closed explanation is still missing (Broecker and Peng (1986); Heinze et al. (1991); Heinze and Hasselmann (1993); Archer and Maier-Reimer (1994); Sigman and Boyle (2000); Archer et al. (2000); Brovkin et al. (2007)). An understanding of the positive feedback be-

tween climate and atmospheric $CO_2$ content (cf. Shakun et al. (2012)) is also important for quantifying the changing carbon cycle today. The glacial-interglacial $CO_2$ variations are mainly caused by a redistribution of carbon between the Earth system reservoirs - ocean, atmosphere, and land. In contrast, the modern carbon cycle is perturbed by the addition of $CO_2$ to the Earth's surface reservoirs. This additional $CO_2$ was previously unavailable to interact with the biogeochemical processes

occurring in these reservoirs. Nevertheless, a reliable reconstruction of glacial-interglacial carbon dynamics with Earth system models would enhance their predictability of future carbon cycle changes. The human-induced $CO_2$ emissions from fossil-fuel burning, cement manufacturing, and land-use change have raised the atmospheric $CO_2$ mixing ratio to 400 ppm relative to a pre-industrial level of 278 ppm (Le Quéré et al. (2014)). The atmospheric $CO_2$ mixing ratio would be considerably higher, if the ocean did not take up a part of the human-made excess $CO_2$ from the atmosphere, primarily through physico-chemical

buffering (Bolin and Eriksson (1959); Maier-Reimer and Hasselmann (1987); Joos et al. (2013)). This marine $CO_2$ buffering has the side effect of reducing the seawater pH (Haugan and Drange (1996); Caldeira and Wickett (2003)). The impact of this progressing ocean acidification on marine biota, ecosystems, and biogeochemical processes is subject to broad interdisciplinary research and harmful effects have been identified (Raven (2005); Gattuso and Hansson (2011)). In particular, a reduction in the formation of calcium carbonate ($CaCO_3$) shell material is key in ocean acidification impact research (Riebesell et al. (2007);

Iglesias-Rodriguez et al. (2008)). Several potential geologic analogues for ocean acidification have been suggested, including: the Paleocene-Eocene Thermal Maximum at 56 Ma (Zachos et al. (2005)), $CO_2$ release from the Siberian Traps Large Igneous Province (Svensen et al. (2009)), and methane release from magmatic intrusion into coal beds (Retallack (2008)). The lack of sedimentary information prior to 180 million years before present limits our knowledge about ecosystems back in deep time. Another prominent example of significant carbon fluctuations is the last glacial-interglacial cycle. For example, the lower $CO_2$

partial pressure at the last glacial maximum (LGM) around 21 kyr BP (21,000 years before present) - could be used as a reverse paleo-analogue of a high $CO_2$ world. Can we say anything about the average behaviour of marine biota with respect to $CaCO_3$ formation under glacial conditions in order to learn about possible future developments? Evidence for better preservation of foraminifera shells is provided by Barker and Elderfield (2002) which would be in line with higher glacial carbonate ion concentrations in surface waters (Yu et al. (2014)).

In this study, we combine a coarse resolution biogeochemical ocean general circulation model (BOGCM) together with long sediment core time series records in order to determine the most likely change in ocean carbon cycle parameters during the last climatic cycle. We also include the "rain ratio" $CaCO_3$:Corg (carbon atoms bound to $CaCO_3$ shell material versus carbon atoms bound to organic carbon in biogenic particulate matter during biological production) in the analysis. This approach would allow us to assess whether the $CaCO_3$ formation in the glacial low $CO_2$ world increased or not. A $CaCO_3$ production

increase at the LGM would be consistent with a decrease of $CaCO_3$ formation in a high $CO_2$ world. The model simulates a time dependent sediment record at each model grid point for variations of single parameters over the past 130 kyr (see section 2). We determine the resulting model sensitivities for the sediment core tracers in relation to a number of key governing parameters of the ocean carbon cycle (see section 3). The sensitivities are projected onto a linear response model. This simplified model is then used for a simultaneous fit of all carbon cycle parameters to available sediment core data from the observed paleo-climatic

archive (section 4). We pursue a multi-tracer approach by considering data on marine $CaCO_3$ wt-%, BSi wt-% (BSi is biogenic

silica or opal, rf. Ragueneau et al. (2000)), $\delta^{13}C_{planktonic}$, $\delta^{13}C_{benthic}$, and the atmospheric $CO_2$ record of the Vostok ice core (Barnola et al. (1987); the EPICA Dome C ice core data will be used to define the sensitivity experiments which are used as input to the linear response model, see below).

## 2 Model description

Here we use the Hamburg Oceanic Carbon Cycle Model HAMOCC (Maier-Reimer (1993); Maier-Reimer et al. (2005)), in its annually averaged version ("HAMOCC2s") in order to keep the computational effort for equilibrating the sediment coverage within feasible limits (Heinze and Maier-Reimer (1999); Heinze et al. (1999); Heinze et al. (2003); Heinze et al. (2009)). This biogeochemical ocean model requires three-dimensional velocity and thermohaline fields as well as ice cover data as input. For the spin-up of the model, a velocity field representing the pre-industrial conditions ("interglacial first guess", Winguth et al.

(1999)) with an annually averaged circulation from the Large Scale Geostrophic dynamical ocean general circulation model (Maier-Reimer et al. (1993)) is employed. The effect of deep convective mixing at high latitudes is considered in this forcing, which is used for transporting the passive tracer substances within the model water column reservoir. The model has a horizontal resolution of $3.5^o \times 3.5^o$, while the vertical resolution decreases downward with 11 layers centred at 25, 75, 150, 250, 450, 700, 1000, 2000, 3000, 4000, and 5000 m. The bioturbated top sediment zone (the "sediment mixed layer") is structured into 10

layers which are separated by "downcore" interfaces at 0, 0.3, 0.6, 1.1, 1.6, 2.1, 3.1, 4.1, 5.1, 7.55, and 10 cm. The simplifying assumption is made that no pore water reactions take place below 10 cm depth in the sediment (following Smith and Rabouille (2002); Boudreau (1997)). HAMOCC2s parameterises the processes of air-sea gas exchange, biogenic particle export production out of the ocean surface layer, particle flux through the water column and particle degradation by remineralisation and re-dissolution, advection as well as mixing of dissolved substances within the ocean velocity field, deposition of particulate

matter on the sea floor, pore water chemistry and diffusion, advection of solid sediment components, bioturbation, and sediment accumulation (export out of the sediment mixed layer or "burial"). The general concept of the sediment model closely follows Archer et al. (1993).

     For the present model configuration, the following tracer concentrations are used as prognostic variables: the atmosphere includes $^{12}CO_2$ (carbon dioxide), $\delta^{13}CO_2$, and $O_2$; the water column contains DIC (dissolved inorganic carbon), alkalinity

(TAlk), POC (particulate organic carbon), DOC (dissolved organic carbon), $CaCO_3$ (calcium carbonate or particulate inorganic carbon) of $^{12}C$ and $^{13}C$, particulate organic phosphorus (POP), dissolved $O_2$, dissolved $PO_4^{3-}$ as a biolimiting nutrient, silicic acid $Si(OH)_4$ and opal (biogenic particulate silica BSi); and sediment processes include pore waters - the same dissolved substances as in the water column - as well as solid sediment with clay, $CaCO_3$, opal, organic carbon, and organic phosphorus.

     The model has been described in detail before (Heinze et al. (1999); Heinze et al. (2003); Heinze et al. (2006)). Details about

the process parameterisations are summarised in Appendix A. In contrast to previous model versions, the treatment of the inorganic carbon chemistry was updated following the procedure as detailed in Dickson et al. (2007). As compared to the previously used quantifications for the reaction constants for the dissociation of carbonic and boric acid according to Mehrbach et al. (1973), the solubility product for $CaCO_3$ after Ingle (1975), and the pressure dependencies of Edmond and Gieskes (1970),

the model required only minor retuning in order to correctly predict the pre-industrial atmospheric pCO$_2$ of ca. 280 $\mu$atm. As before, DIC and TAlk are used as "master tracers" from which the other derived inorganic carbon species such as the CO$_3^{2-}$ concentration, the carbonate saturation, and the pH value are computed through a Newton-Raphson algorithm. A further extension of the model as presented in Heinze et al. (2009) is the introduction of separate fields for POP (particulate organic phosphorus), DOP (dissolved organic phosphorus), and solid organic phosphorous in the sediment mixed layer to allow for variable stoichiometry ("Redfield ratios" C:P) in this study. The other ocean water column and pore water processes are represented through parameterisations as described for in earlier model versions

## 2.1   Transport of age information and passive tracers in the sediment

Next to the various solid sediment components, the model also predicts the age of each sediment component. The age of the amount of each component rained onto the model seafloor is set to the actual model time step. The age information is then handed down into the model sediment layer as a passive tracer. This information transport is also dependent on the pore water dissolution reactions, the deposition flux, the vertical sediment advection, and the bioturbation. Each solid sediment species has its own age tracer. The higher the sediment accumulation rate is the closer the age values of the four different sediment component ages are. For a schematic of the sediment model, see Heinze et al. (2009) (Figures 1 and 2 therein).

The model sediment is initialised at age 0 with only clay sediment layers and then spun-up. During the progressing integration, the model builds up its own biogenic sediment and clay sediment according to the dust deposition and the kinetic control of the continental inputs of nutrients, carbon, and alkalinity as prescribed through bulk numbers. Simultaneously with the sediment accumulation, the age structure of each solid weight fraction is established. In equilibrium, the global sediment accumulation rates are a function of the continental input rates. The local sediment accumulation rates and sediment compositions are fully prognostic variables. In order to also "flush" the sediment mixed layer in low accumulation regions of the world ocean, the model is spun-up for 100,000 years to achieve overall quasi-equilibrium (as documented by the stability of the atmospheric pCO$_2$ level and the asymptotic approach of the simulated globally integrated sediment accumulation rates to the globally prescribed continental matter input fluxes).

The model includes the cycling of $^{13}$C. The observed foraminiferal CaCO$_3$ sediment record reflects $\delta^{13}$C variations from both the surface and the deep waters (depending on the shell material analysis of the different surface and deep dwelling species). As an extension to Heinze et al. (2009), the present model version also includes sedimentary benthic $\delta^{13}$C. Normally, only the surface ocean $\delta^{13}$C ($\delta^{13}$C$_{planktonic}$) is included in the simulated CaCO$_3$ deposited onto the model top sediment layer and then treated subsequently during pore water reactions and sediment advection. The reason for this is that we have no explicit benthic production of foraminifera in the model. Applying the basic concept of the passive age tracer treatment in the early diagenesis module, we also transport the $\delta^{13}$C$_{benthic}$ signal of the CaCO$_3$ fraction within the sediment mixed layer (where we set the benthic solid $\delta^{13}$C value for the depositing CaCO$_3$ material to the respective bottom water $\delta^{13}$C of the model layer directly over the respective top sediment box). Thus, the following sedimentary variables are included in the present model for direct comparison with values from sediment core analysis: the "foraminiferal" values of $\delta^{13}$C$_{planktonic}$, $\delta^{13}$C$_{benthic}$ (as the latter would be recorded in reality, e.g., by the foraminiferal species *Cibicidoides wuellerstorfi*, cf. Zahn et al. (1986)),

and also their vertical difference $\Delta\delta^{13}C=\delta^{13}C_{planktonic}-\delta^{13}C_{benthic}$ as an indicator of the strength of the biological pump (or the vertical nutrient gradient in the water column, see Shackleton and Pisias (1985). The model master tracers DIC and TAlk for the inorganic carbon cycle include the sum of $^{12}C$ and $^{13}C$. Explicit $^{13}C$ concentrations are computed from the spin-up through the requirement that the pre-industrial atmospheric $\delta^{13}C$ has a value of -6.5 ‰. The continental input rate of $^{13}C$ and the corresponding output rate through sediment accumulation where iteratively determined, so that in the final model spin-up the $^{12}C$ and $^{13}C$ are fully compatible with the pre-industrial $\delta^{13}C$ value.

## 2.2  Control run

Important global bulk numbers for the quasi-equilibrium state of the 100,000 yr long control run are given in Table 1. For illustration of the control performance, examples for the export production rates and the respective sediment components are provided in Figure 1. Meridional cross sections for dissolved inorganic phosphate, carbonate ion concentration, and $\delta^{13}C$ of total dissolved inorganic carbon are shown in Figure 2. The model is able to reproduce the major characteristics of global tracer distributions, with decreasing deep water carbonate ion concentrations and $\delta^{13}$ values as well as increasing phosphate concentrations from the Atlantic to the Pacific Ocean. Simulated tracer patterns appear to be more smoothed as compared to the observations because of the coarse spatial resolution of the model. The $CaCO_3$ sediment distribution in the northern Pacific Ocean may show somewhat too high $CaCO_3$ wt-% as compared with observations. Based on experience with previous simulations, the $CaCO_3$ content of the sediment mixed layer appears to react quite sensitively to the degree of undersaturation and the redissolution rate constants chosen. Therefore, a slight overestimate of $CaCO_3$ sediment in the control run avoids a potential complete dissolution of $CaCO_3$ for many grid points during the sensitivity experiments.

## 3  Sensitivity experiments

In order to determine, how the governing process parameters of the carbon cycle changed from the cold low $CO_2$ world to the warmer and higher $CO_2$ pre-industrial world, we first carried out a suite of sensitivity experiments with the full 3-D carbon cycle model. In each run, we change only one of the governing parameters in order to determine the respective change of the biogeochemical system. In particular, we are interested in the respective change of the inorganic as well as organic carbon cycle and how these changes are imprinted onto the simulated paleo-climatic record. In a previous study Heinze et al. (1991) carried out sensitivity experiments with an earlier version of HAMOCC and also compared the results with data from the observational paleo-climatic archive. The present study, however, differs significantly from the previous approach. In Heinze et al. (1991) only equilibrium responses of the biogeochemical state to instantaneous parameter changes (permanent switch from one of the control run configurations) were tested, the alternative velocity field used did not resemble the glacial ocean very well, and the model did not include an interactive early diagenesis sub-model (only interactive bulk sediment reservoirs which did not allow for varying total system inventories). The latter issue thus allowed only a comparison between the modelled tracer values and sediment core derived "proxy data". Also the fixed total system inventory of carbon, nutrients, and alkalinity allowed a limited range of system changes. The first study of HAMOCC including an interactive diagenesis module by Archer and Maier-Reimer

(1994) already revealed the higher sensitivity of such a model configuration, as the water column tracer inventory could vary considerably more dynamically.

We first carried out a control run over the past 130,000 years, re-starting from the standard spin-up of the model. This control run revealed the excellent mass conservation and also the almost drift-free $^{13}$C distributions in all reservoirs. Thereafter, a series of simulations with perturbed model parameters were conducted by restarting from the same pre-industrial control run. The runs were started from "Eemian" conditions which we assumed to be similar to the pre-industrial state as resulting from the HAMOCC spin-up. We then scaled the various parameter perturbations with the $\delta$D temperature record of the EPICA Dome C ice core project (Jouzel (2004)) assuming that all carbon cycle variations during the past climatic cycle are either correlated with temperature or with the rather similar temporal pattern of the atmospheric $CO_2$ variation. There are uncertainties associated with this assumption, as the Antarctic air temperature record is a localised record and may not reflect global temperature change, and also because ocean temperatures may lag behind this signal. On the other hand, the local Antarctic $\delta$D temperature record is highly correlated with the global atmospheric p$CO_2$ signal (Siegenthaler et al. (2005); Barnola et al. (1987); Jouzel et al. (1987)). The seawater temperature change would probably not lag too long behind this signal if reasonably long time intervals are considered. Due to the close correlation between air temperature and atmospheric p$CO_2$ over the past glacial-interglacial cycle, we cannot strictly discriminate between drivers for ocean and land carbon cycle changes coming from physical (temperature) and biogeochemical forcings (p$CO_2$, carbonate saturation, pH value), but we need to justify this independently for the different parameters under investigation. For the sensitivity experiments, we selected the following governing model parameters of the carbon cycle: (1) The sea surface temperature for computation of the chemical and biological constants, (2) the release/uptake of carbon from/by the terrestrial biosphere, (3) the degradation rate constant of POC in the water column, (4) the dissolution rate constant of BSi in the water column, (5) the export production rain ratio $CaCO_3$:POC, (6) the 3-D oceanic velocity field, (7) the glacial dust deposition and associated stimulation of biological export production, and (8) the Redfield ratio C:P.

These parameter changes are summarised in Figure 3 and Table 2 and described in more detail below.

## 3.1   Sea surface temperature for computation of the chemical and biological constants

In this experiment, we reduced the sea surface temperature up to a maximum change of 5 K (with the consideration that the minimum temperature stays at or above the freezing point of sea water) only for the computation of the temperature dependent chemical and biological parameters in the model, which usually all have their fixed control run values. The effect on the $CO_2$ solubility is relatively strong, accounting for ca. 12 ppm change in atmospheric p$CO_2$ for a 1 K change in sea water temperature (cf. Broecker and Peng (1986)). The 5 K change in temperature is approximately the maximum change that can be expected on the basis of stable oxygen isotopes or other temperature indicators (e.g., Stute et al. (1995)).

## 3.2   Release/uptake of carbon from/by the terrestrial biosphere

Variations in the land carbon cycle also need to be properly taken into account for the overall p$CO_2$ signal in the atmosphere and the imprint of $^{13}$C on the sedimentary record. We do not explicitly model the terrestrial carbon cycle in our experiment. Rather

we assume here, that the land biosphere would have a smaller net primary production (Hoogakker et al. (2016)) and standing biomass stock and hence provide a release of carbon to the atmosphere under colder and dryer conditions. The amplitude for biomass carbon loss from the land biosphere to the atmosphere was set to -1000 PgC, in line with estimates from terrestrial paleo-climate records according to Crowley (1995). A potentially increased storage of organic carbon under the ice sheets and

at cold conditions (lower bacterial activity for plant biomass degradation) is also being discussed (e.g., Zeng (2003)). Such storage could lower the overall carbon loss from the terrestrial biosphere under glacial conditions.

### 3.3   Dissolution rate constants of POC and BSi

We explored the temperature effect on both sinking particulate organic carbon as well as biogenic silica. We assumed that the degradation of both substances slows down under lower temperatures due to thermodynamic effects as well as decreasing bac-

terial decomposition (Turley and Mackie (1995); Van Cappellen et al. (2002), Bidle and Azam (1999)). Through the respective parameter changes we tested separately a strengthening of the biological carbon pump and of the silicon pump which would have led to a deeper penetration of carbon and nutrients into the water column under glacial conditions ("vertical fractionation" as described by Boyle (1988a) and Boyle (1988b)). We chose a maximum decrease of degradation strength of 10%.

### 3.4   Export production rain ratio CaCO$_3$:POC

A decrease in rain ratio has been suggested as one of the potential mechanisms to extract carbon from the atmosphere during glacial times (Berger and Keir (1984)). A reduction of the CaCO$_3$ production to zero could potentially explain half of the glacial pCO$_2$ drawdown from the atmosphere (Broecker and Peng (1986)). However, the experiments by Zondervan et al. (2001) and Riebesell et al. (2000) would suggest the opposite: in a low CO$_2$ world, the CaCO$_3$:POC ratio would be expected to increase. This would make it more difficult to explain the glacial CO$_2$ uptake by the oceans. Heinze and Hasselmann (1993) showed

that this is possible in principle. In our experiment here, however, we follow the idea of a diminished CaCO$_3$:POC export production rain ratio during glacial times. In order to get a pronounced signal, we decrease the amplitude to a -10% change at the LGM with respect to the control run.

### 3.5   3-D oceanic velocity field

Practically all geochemically relevant paleoclimate tracers in the ocean depend on the ocean circulation. Therefore, it is nec-

essary to take a realistic glacial velocity field into account in order to provide the correct addition to tracer changes which may be induced by biological and chemical processes. Here, we follow the same approach as outlined in Heinze (2001), where we "mix" the velocity, thermohaline, convective mixing, as well as sea ice margin values from two different runs of the dynamical ocean general circulation model LSG (Large Scale Geostrophic Model, Maier-Reimer et al. (1993)). These runs are the respective interglacial and glacial first guess runs from Winguth et al. (1999) which show major features of the pre-industrial

as well as LGM distributions of the $\delta^{13}$C of dissolved inorganic carbon. As maximum amplitude for the velocity field change we assume a full switch to LGM conditions at the largest negative excursion of the $\delta$D EPICA Dome C ice core curve. It would

be preferable to have respective atmospheric forcing fields available to have a better constraint on the velocity fields at hand. However, the considerable effort required to realise such a run (which would need to be carried out with the full seasonally resolving LSG model) was deemed too large for our study here. The "kinematic" approach chosen here should work for practical reasons as the respective $\delta^{13}$C distributions of DIC show realistic patterns.

## 3.6 Glacial dust deposition

We carried out a number of sensitivity tests concerning the effect of a change in continental dust deposition onto the sea surface. First we only tested the pure effect of clay addition through dust by applying the glacial dust deposition field by Mahowald et al. (1999) at the LGM and respective mixtures of interglacial and glacial dust deposition fields at other time intervals. As the dust records of marine sediment core records usually do not follow the pattern of the $\delta$D record (e.g., Figure 4 of Carter and Manighetti (2006)), we scaled the change in dust supply with the $6^{th}$ order of the glacial-interglacial difference of the $\delta$D curve, i.e. assumed a sudden increase of dustiness only at really low temperatures. Again we set a maximum value of dust change to 100% glacial conditions at the LGM and smaller changes elsewhere (see Figure 3). The effect of this dust addition would be a change in the dilution of the sediment weight percentages of the reactive sedimentary material by inert clay and also a change in the local sediment accumulation rates. It has been suggested that a stimulation of biological export production through increased dust and iron deposition to the ocean surface under glacial conditions could occur (e.g., Martin et al. (1994); Berger and Wefer (1991)). In an additional experiment, the maximum uptake velocities for phosphorus, carbon, and silica were scaled with the glacial-interglacial difference of the dust deposition as provided by Mahowald et al. (1999). For the results of the fitting procedure as shown below, we considered here the run with the dilution effect only in order to separate sediment dilution effects on the CaCO$_3$ sediment concentration from those caused by an increase in biological production.

## 3.7 Redfield ratio C:P

Finally we allowed carbon over- or underconsumption in response to the interglacial to glacial change in environmental conditions. Riebesell et al. (2007) and Bellerby et al. (2008) postulated an increasing carbon overconsumption C:N under high CO$_2$ conditions resulting from a mesocosm experiment. Conversely, a change towards increasing carbon overconsumption during the glacial low CO$_2$ world had been suggested as a powerful mechanism to account for the glacial CO$_2$ drawdown and at the same time cause a plausible foraminiferal $\delta^{13}$C signal (e.g., Shackleton and Pisias (1985); Broecker (1982); Broecker and Peng (1986); Heinze and Hasselmann (1993)). In our sensitivity experiment we investigated the possibility for an increase in carbon overconsumption by imposing an increase in the Redfield ratio C:P homogeneously over the ocean. We specified a maximum change of 15% at the LGM (see Figure 3, lowermost panel).

We try to make a reasonable trade-off between the number of free parameters and the need for inclusion of the most essential parameters which contribute to the tracer variations in the paleo-record. Thus we aim to isolate imprints of parameter variations in the observed sedimentary record which would otherwise be masked by further independent processes (e.g., it would be futile to derive changes in biological pump strength from the foraminiferal $\delta^{13}$C distribution in the ocean without taking into account respective changes in the ocean currents which also affect the marine $\delta^{13}$C distribution).

For determining the most likely choice of the governing parameters, we analysed modelled output for a series of paleo-climatic data sets. The model delivers first of all prognostic atmospheric $pCO_2$ values for each sensitivity experiment relative to the control run during each time step for comparison with the Antarctic ice core records. The model runs were started from year 129,536 BP and integrated until the present. This start year was chosen so that the first and last $\delta D$ values are close to each other and no anthropogenic $CO_2$ emissions were imposed during the very last part of the integration. Each run requires around 4 days CPU time on an advanced UNIX work station. Using the methodology of Heinze et al. (2009) the model at each grid point accumulates sediment in a continuous way for the components $CaCO_3$, opal, organic carbon, clay, and organic phosphorus. $\delta^{13}C_{planktonic}$ and $\delta^{13}C_{benthic}$ of the $CaCO_3$ fraction and the individual ages of each sediment component plus the atmospheric $CO_2$ partial pressure were stored together with the actual accumulation rates at the base of the sediment mixed layer. In order to reduce the amount of data output, we only stored 100-yr averages. Over the integration time period, for each run a continuous temporally varying sediment stratigraphy is built up. After the model runs, we "drill" into this synthetic sediment and "recover" simulated sediment cores for comparison with the observations from real world sediment core data. We can provide such modelled cores at every grid point of the model individually.

## 4 Linear response model, observational sediment core data base, and fitting procedure

The inverse modelling procedure as applied and described here, draws on many aspects of the work by Heinze and Hasselmann (1993). We partly repeat some of the methodological issues here, to avoid the reader having to refer too often to a separate study when reading this analysis here. The method is illustrated in Figure 4. We carried out a total of eight sensitivity experiments with the full 3-D BOGCM, providing 8 different data sets for respective changes in 79 different tracer time series (see below) resulting formally in a total of 20,540 modelled tracer values and the same number of respective observed values (for each run). The eight sensitivity experiments represent the response of the full 3-D model to variations of $n=8$ governing input parameters $x_j$ ($j=1, \ldots, n$) (Table 2) resulting in $m=20,540$ paleoclimate tracer record changes $g_i$ ($i=1, \ldots, m$) as induced in these experiments as a consequence to the changes $x_j$ of the parameters.

The resulting linear response model thus consists of the matrix describing the linearised relation between perturbations of the input and output variables. The model output variables $\hat{g}_i$ in the sensitivity experiments correspond to the observed perturbations of the atmospheric $CO_2$ record and the sediment records of $CaCO_3$, $\delta^{13}C_{benthic}$, $\delta^{13}C_{planktonic}$, and opal (BSi) (see following sub-section).

In the full non-linear BOGCM, the paleo-climate tracers $G_i$ are functions of the governing model parameters $X_j$,

$$G_i \quad = \quad F_i(X_j) \quad . \tag{1}$$

In the simplified linear response model, the relation (1) is Taylor expanded (and then truncated) about a reference state $X_j = X_j^0$, $G_i = G_i^0$ as given by the standard run (with zero parameter changes):

$$\hat{g}_i \quad = \quad \sum_{j=1}^{n} A_{ij} x_j \tag{2}$$

where $\hat{g}_i = G_i - G_i^0$, $x_j = X_j - X_j^0$ and $A_{ij} = \frac{\partial F_i}{\partial X_j}(X_j^0)$. The response coefficients $A_{ij}$ form the elements of the model matrix $A$. These response coefficients resulted from the model sensitivity experiments under the assumption of a linear relation between the parameter changes $x_j$ and the response vector $\hat{g}_i$ predicted in every sensitivity experiment $j$.

## 4.1 Data base of observations from the paleo climate record

For calibrating the free model parameters, we employ a data base of paleo-climatic records as summarised in Tables 3 and 4. Locations are indicated in Figure 5. Most of the $\delta^{13}C_{benthic}$ and $\delta^{13}C_{planktonic}$ data were taken from the compilation of Oliver et al. (2010). The majority of $CaCO_3$ wt-% data were taken from a compilation by Hoogakker (pers. comm., Table 4). Further marine sediment core data ($\delta^{13}C_{benthic}$, $\delta^{13}C_{planktonic}$, BSi, and $CaCO_3$ wt-% ) were taken from various literature sources (see references in Tables 3 and 4). For atmospheric $pCO_2$ we use the Vostok ice core signal from Antarctica

(Barnola et al. (1987)). We did not explicitly synchronise the different paleo-climatic time series to a common age model here, but rather took the measurement/age combinations at face value. We estimate the error due to this to up to a few thousand years (compare the discussion in Oliver et al. (2010)). As we are not interested here in exact timing and phase shifts of signals in different variables, and due to the overall errors in the modelling approach (coarse resolution model, only crude representation of circulation changes, assumption that pre-industrial model state corresponds also to the Eemian) we assume that this approach

is reasonable. Some sediment core data were extracted even from published graphics by hand. Most of the $\delta^{13}C_{benthic}$ and $CaCO_3$ wt-% time series employed, however, were synchronised in respective data compilations. For the fitting procedure, we interpolated all observed data to regular time series with an increment of 500 years (260 equidistant data points in time).

  Modelled annual global mean values of atmospheric $pCO_2$ were compared for comparison with the Vostok $pCO_2$ signal. The corresponding model time series data of the different sediment paleoclimate tracer curves were extracted from the respective

model grid point closest to the location of real sediment core extraction for the various model sensitivity experiment runs. Modelled sediment data were interpolated with respect to age onto the same 500 years points as the observations. The age of each sediment component was simulated according to Heinze et al. (2009) during the sensitivity experiments and stored together with the simulated depth in core of each modelled sediment variable.

  For the fitting procedure, observed and modelled data where translated to changes relative to the preindustrial. Thus, we

analysed the changes in the tracers relative to this reference while the respective parameter changes were taken as changes relative to the control run values.

## 4.2 Fitting procedure

Our general fitting procedure closely follows the method as described in Heinze and Hasselmann (1993)).

### 4.2.1 The general minimisation problem

We fitted the linearly modelled paleoclimatic tracer changes $\hat{g}_i$ to the observed tracer changes $g_i$ through minimizing the mean square sum $Q^2$ of the individual tracer errors $\epsilon_i = \hat{g}_i - g_i$:

$$Q^2 \quad = \quad \sum_{i=1}^{m} \epsilon_i^2 \quad = \quad \sum_{i=1}^{m} (\sum_{j=1}^{n} A_{ij}\, x_j \,-\, g_i\,)^2 \tag{3}$$

$$= \quad (\,\cdot x^T{}_| \,\,{}_|A^T| \,-\, \cdot g^T|\,)\,(\,|A_|\,\,{}_|x\cdot \,-\, |g\cdot\,)$$

The dimensions of the vectors and matrices are described by the following symbols (cf. Heinze and Hasselmann (1993)): The left marker of a matrix symbol denotes the number of rows, the right marker specifies the number of columns. In our specific case, the marker "$\cdot$" corresponds to 1, "$|$" to the number $m = 20,540$ of observed tracer data, and "$_|$" to the number $n = 8$ of parameter changes $x_j$ to be estimated. The solution $_|\hat{x}\cdot$ which minimizes the mean square expression $Q^2$ is derived through the necessary condition $\frac{dQ^2}{dx} = 0$ :

$$_|A^T|\,\,{}_|A_|\,\,{}_|\hat{x}\cdot \quad = \quad {}_|A^T|\,\,|g\cdot$$

which can also be written as

$$_|\hat{x}\cdot \quad = \quad (\,{}_|A^T|\,\,{}_|A_|\,)^{-1}\,{}_|A^T|\,\,|g\cdot \tag{4}$$

Through one realisation of the fitting procedure we thus, simultaneously, determined all eight optimal parameter changes $\hat{x}_j(t)$ necessary for a best possible reproduction of the observed tracer changes. In addition, the procedure yields a time series of residual errors $\epsilon_i(t)$ which quantify the expected remaining discrepancies between modelled and observed tracers after the optimisation (we write here "expected" as the non-linear model may result in somewhat different residuals than the linear response model). Our problem is formally overdetermined where $m = 20,540$ tracer data points must be fitted with $n = 8$ parameters. Therefore, the least squares formulation of our solution procedure is appropriate.

The error expression of eq. (3) is defined with a simple unit matrix norm. The more general maximum likelihood norm would be given by the inverse of the covariance matrix of the measurement errors (e.g., Martin (1971)). In eq. (3), we make the assumption that the errors are uncorrelated and have the same variance. At present, we have no conclusive means for estimating the error correlation. Therefore, the assumption of a simple unit matrix norm is reasonable here. However, we normalised the variables $g_i$ by the mean absolute values of the respective tracer over the past climatic cycle, so that differential weighting of different tracer records, e.g., already from the use of different physical units is smoothed out.

### 4.2.2 The SVD least squares solution

In many inverse problems it is not possible to determine a physically meaningful least squares solution directly from eq. (4). The solution can be contaminated by noise if the model matrix $|A_|$ is badly conditioned. This leads, in practice, to unrealistically high variations of the optimised parameter changes if these depend only on very small variations in the tracer changes. Beforehand, it cannot be seen whether $A$ is badly conditioned or not. Therefore, we applied the singular value

decomposition (SVD) technique which provides a quantitative treatment of the noise problem. For an introduction to this technique, please, see the very useful work by Wunsch (1989). The model matrix $A$ is decomposed into a product of three matrices (SVD; see Lanczos (1961)):

$$|A| \quad = \quad |U| \ |\Lambda| \ |V^T| \tag{5}$$

The column vectors of $|U|$ and $|V|$ represent orthonormal bases $|u_i\cdot$, $i = 1,\ldots,m$ and $|v_j\cdot$, $j = 1,\ldots,n$ for the tracer (here tracer change) and parameter (here parameter change) spaces, respectively, associated with the model matrix $|A|$. $|\Lambda|$ The diagonal rectangular matrix

$$|\Lambda| \quad = \quad \begin{pmatrix} \lambda_1 & 0 & . & . & . & 0 \\ 0 & \lambda_2 & . & . & . & . \\ . & . & . & . & . & . \\ . & . & . & . & . & . \\ . & . & . & . & . & . \\ . & . & . & . & . & 0 \\ 0 & . & . & . & 0 & \lambda_n \\ 0 & . & . & . & . & 0 \\ . & . & . & . & . & . \\ 0_{1,m} & . & . & . & . & 0_{n,m} \end{pmatrix} .$$

consists of the singular values $\lambda_k$. The eigenvectors $|u_k\cdot$, $|v_k\cdot$ and the eigenvalues $\lambda_k$ form the solution of the coupled eigenvalue

problem:

$$|A| \ |v_k\cdot \quad = \quad \lambda_k \ |u_k\cdot \tag{6}$$

$$|A^T| \ |u_k\cdot \quad = \quad \lambda_k \ |v_k\cdot \tag{7}$$

with the following properties:

$$|A| \ |A^T| \ |u_k\cdot \quad = \quad \lambda_k^2 \ |u_k\cdot \tag{8}$$

$$|A^T| \ |A| \ |v_k\cdot \quad = \quad \lambda_k^2 \ |v_k\cdot \ . \tag{9}$$

The number $p \leq min(m,n)$ of non-zero singular values defines the rank of $|A|$.

The unknown parameter changes $|\hat{x}\cdot$ and the modelled tracer changes $|\hat{g}\cdot$ can now be written in terms of the linearly independent eigenvectors $|v_j\cdot$ and $|u_i\cdot$, respectively:

$$|\hat{x}\cdot \quad = \quad \sum_{j=1}^{p=n<m} \alpha_j \ |v_j\cdot \tag{10}$$

$$|\hat{g}\cdot \quad = (g_1,\ldots,g_m) = \sum_{i=1}^{p=n<m} \beta_i \ |u_i\cdot \tag{11}$$

where the $\alpha_j$ are unknown and $\beta_i = \cdot u_i^T| \ |g\cdot$. The observed tracer changes are correspondingly given by $|g\cdot \equiv \sum_{i=1}^{m} \beta_i \ |u_i \cdot)$ Equations (4), (7), and (9) together yield the solution for the optimal parameter changes:

$$|\hat{x}\cdot \ = \ \sum_{j=1}^{n} \alpha_j \ |v_j \cdot \ = \ \sum_{i=1}^{n} \frac{1}{\lambda_i} |v_i\cdot \ \cdot u_i^T| \ |g\cdot \ . \tag{12}$$

with $\alpha_i = \beta_i/\lambda_i$. Because only $p = n < m$ vectors $|u_i\cdot$ are combined in the overdetermined case, residuals between mod-elled tracer changes $|\hat{g}\cdot$ and observed tracer changes $|g\cdot$ occur. Thus, the observed tracer variations cannot be completely recombined by the least squares solution of eq. (12).

In the exactly determined case we would have

$$|\hat{g}\cdot \ \equiv \ |g\cdot \ = \ \sum_{i=1}^{m} \beta_i \ |u_i \cdot \ = \ \sum_{i=1}^{m} |u_i\cdot \ \cdot u_i^T| \ |g\cdot \ = \ |(\delta_{ij})| \ |g\cdot \tag{13}$$

where $\delta_{ij} = \begin{cases} 1 & for \quad i=j \\ 0 & for \quad i\neq j \end{cases}$.

For the overdetermined case we arrive at

$$|\hat{g}\cdot \ = \ \sum_{i=1}^{p=n<m} \beta_i \ |u_i \cdot \ = \ \sum_{i=1}^{p=n<m} |u_i\cdot \ \cdot u_i^T| \ |g\cdot \ = \ |\hat{U}| \ |\hat{U}^T| \ |g\cdot \ = \ |R_I| \ |g\cdot \tag{14}$$

with $|R_I| \ = \ |\hat{U}| \ |\hat{U}^T|$ being the "resolution matrix" (Wiggins (1972); Wunsch (1989)) of the changes in the tracer concentra-tions and $|\hat{U}| \ = \ ( \ |u_1\cdot , \ldots, \ |u_n\cdot )$ is the truncated "model" version of the tracer base matrix $|U|$.

According to eq. (12), the coefficient $\alpha_i$ in the linear combination of the parameter space eigenvectors $|v_i \cdot$ grows to very large values if $\lambda_i$ becomes very small. These very small $\lambda_i$ values lead to an unstable solution: only unreasonably large changes in selected parameters would be able to create the tracer changes as observed. The contributions to the solution from these small eigenvalues should be filtered out accordingly or considered with a significantly reduced weight.

For the resulting parameter changes from our fitting procedure, we discarded those components in eq. (12) associated with the smallest singular values step by step starting with the smallest singular value, followed by the next smallest one, and so on to filter out unrealistically large parameter changes. The associated approximate solution then becomes

$$|\hat{x}\cdot \ = \ \sum_{i=1}^{p<n} \frac{1}{\lambda_i} |v_i\cdot \ \cdot u_i^T| \ |g\cdot \tag{15}$$

The number $p$ of eigenvalues retained is called the effective rank of $|A|$. In our case here the full rank solution would be achieved for a rank=8 (including all parameter changes). For an effective rank smaller than $min(m,n)$, the formally overde-termined problem is now changed into an underdetermined problem. Nevertheless solution eq. (15) is the unique least squares solution with minimum length of the solution vector for the underdetermined case (e.g., Matsu'ura and Hirata (1982)). Such a solution with minimum overall change of the system under investigation is often physically or chemically seen to be the most appropriate solution ("Occam's razor").

Because the filtered solution relies on fewer fitted parameters changes, the goodness of fit of this filtered solution is poorer than in the full rank SVD solution. Because the rank $p$ of the reduced matrix is smaller than the number of original param-eters, the fitted parameters are no longer linearly independent. Now, one can specify only $p$ linearly independent parameter

combinations. The full rank solution ($p = n \leq m$) would be:

$$\hat{x}_{full\ rank} \ = \ \sum_{j=1}^{p=n} \alpha_j\, v_j\, = \ \sum_{j=1}^{p=n} v_j \cdot v_j^T\, \hat{x}_{full\ rank} \ = \ (\delta_{ij})\, \hat{x}_{full\ rank} \tag{16}$$

The solution for the rank deficient case ($p < n \leq m$) is now:

$$\hat{x} \ = \ \sum_{j=1}^{p<n} \alpha_j\, v_j\, = \ \sum_{j=1}^{p<n} v_j \cdot v_j^T\, \hat{x}_{full\ rank} \ = \ \hat{V}\, \hat{V}^T\, \hat{x}_{full\ rank}$$

$$= \ R_P\, \hat{x}_{full\ rank}\ . \tag{17}$$

$R_P = \hat{V}\,\hat{V}^T$ is the "resolution matrix" of the parameters with $\hat{V} = (v_1, \ldots, v_p)$. This resolution matrix $R_P$ projects the full (in part insufficiently constrained) set of parameters onto the sub-space of stable solutions for the parameter changes. If for fixed $i$ the diagonal element $R_{ii}^P$ of the matrix $R_P$ is close to 1 and the other elements $R_{ij}^P$, $j \neq i$ are close to zero, parameter $x_i$ is still well resolved even in the underdetermined case. Less well resolved parameters, are instead replaced by a stable linear combination of the original parameters. Through analysis of the resolution matrix for the parameters, we can therefore determine which parameter changes can be uniquely determined by the analysis and for which parameters no clear conclusion can be made.

## 5 Results and their discussion

We carried out numerous runs with the linear response model, testing different combinations of parameters and using different observed time series from paleoclimate records as constraints. Overall we experienced, that the paleoclimatic information and the system sensitivity as provided by the biogeochemical ocean model lead to quite consistent results, if several parameters and paleoclimatic tracers are taken into account.

We present results where all parameter changes and all paleoclimatic tracer curves were included (see summaries in Table 5 and Figures 6-8). The 8 unknowns (namely the excursions of the parameter amplitudes relative to interglacial conditions) where determined by an optimal fit to the 20,540 data points from the observational records. Formally, such a system is heavily overdetermined. No *a priori* knowledge and no artificial limits where imposed to the free parameters to be determined. Are the results for the free parameters reasonable? Figure 6 shows the parameter time series for the full rank solution (compare to the first guess values as shown in Figure 3), which also gives the best fit to the observations. Please, note the model parameter changes by purpose show the same temporal pattern (special case dust as discussed above) and only the maximum amplitude of the parameters was determined. The *a posteriori* parameter changes remain relatively close to the initially chosen parameter changes (first guesses), yet with one exception. The rain ratio change shows a dramatic decrease in pelagic CaCO$_3$ production. Such a change may not be fully out of scope (see, e.g., the discussion in Broecker and Peng (1986) and Berger and Keir (1984)). Archer and Maier-Reimer (1994) argued that enhanced CaCO$_3$ dissolution on the sea floor through organic carbon degradation in combination with a rain ratio reduction would provide an efficient way for reducing atmospheric pCO$_2$. The rain ratio change itself could be provided by an increased surface ocean concentration of silicic acid by which diatoms would dominate

over $CaCO_3$ shell material production. Such a change in silicic acid concentration could be induced by enhanced iron fluxes to the Southern Ocean by dust, thinner opal frustules after the iron stress has been diminished, and subsequent export of "unused" silicic acid from the Southern Ocean to the rest of the world ocean ("silicic acid leakage hypothesis", Matsumoto et al. (2002); Griffiths et al. (2013)). Further it has been argued that low seawater temperatures lead to lower water column reminer-

alisation rates for organic carbon and changes in the ecosystem community structure that would imply a rain ratio reduction (Matsumoto et al. (2007)). On the other hand, in case of a strong coupling between deep POC fluxes to $CaCO_3$ fluxes (where $CaCO_3$ works as ballast for downward organic carbon transport; see Klaas and Archer (2002) and Armstrong et al. (2002)), rain ratio shifts at the ocean surface would only have a minor impact on atmospheric $pCO_2$ (Ridgwell (2003)). Further, one would rather expect an increase in $CaCO_3$ production at low ambient $pCO_2$ and high $CaCO_3$ saturation (Zondervan et al.

(2001); Riebesell et al. (2000)). Therefore, the suggested strong change in the rain ratio may be an unstable part of the solution, where the fitting procedure needs huge changes in one (or possibly several parameters) to achieve minor changes in the paleoclimate tracers simulated. (Nevertheless, the corresponding SVD solution is the unique solution of the system for full rank with altogether minimum deviation from the control run.) We, therefore, started to reduce the rank from 8 to first 7 and then 6, i.e. we still aimed at determining all 8 parameter changes simultaneously, but partially in linear combination with each

other. In the solution for rank 7 (Figure 7), the change in sea surface temperature seems to be unrealistically high. A global decrease by about 8°C would imply an unrealistic widespread ocean freezing. The solution for rank 6 (Figure 8) finally arrives at overall realistic parameter excursions during the past climatic cycle. For maximum changes at the LGM, our analysis arrives at:

- A maximum decrease in global sea surface temperature by 5°,

– a decrease of the land biosphere by 400 PgC,

- only minor changes in POC and BSi degradation,

- a suggested change of the ocean velocities towards glacial conditions (no full switch however),

- only a small effect from dust deposition,

- and a sizable strengthening of the biological organic carbon pump through a stoichiometry change.

The resulting requirement for a strengthening of the biological pump could potentially also be due to other processes than a stoichiometry change in reality.

The suggested glacial drop in average SST by 5°C may still be overestimated. Originally, CLIMAP Project Members (1976) and Broecker (1986) estimated a glacial-interglacial change of only ca. -2.0°C to -2.3°C from faunal assemblages and oxygen isotopes, with the more recent quantification by MARGO Project Members (2009) being at the lower end of this range. Other

estimates suggest a decline in the SST of 5.4°C Stute et al. (1995)) and even 6.5°C (Weyhenmeyer et al. (2000)). Green et al. (2002) arrive at an estimate for tropical SST glacial decrease of around 3°C but give a range of studies (see their Table 2 on p. 4-9 in Greene et al., 2002). Our study confirms the order of magnitude in this parameter, using an independent approach.

A decrease in land vegetation due to colder and drier climate conditions at the LGM relative to pre-industrial conditions has been estimated by several authors employing various approaches: 430 PgC (Shackleton (1977), 530-1160 PgC (Crowley (1995)), 330 PgC (Ciais et al. (2012)), and 597 PgC (O'ishi and Abe-Ouchi (2013)). Interestingly, our estimate is closest to the value originally determined by Shackleton (1977), which may be due to the marine $\delta^{13}$C constraint used in that study and which is also included in our analysis. Overall, our study confirms earlier estimates on the LGM terrestrial through our independent approach.

Our fitting procedure confirms that the glacial ocean circulation should have somehow resembled the simulated glacial ocean conditions as deduced by Winguth et al. (1999). However, in an ideal case with a perfect simulation of the glacial ocean circulation, the switch to this circulation at glacial conditions should be close to 100 % and not only the 50 % seen in this study. This may still be due to some deviations of the simulated glacial ocean velocity especially for the Pacific Ocean, but possibly also due to deficiencies in the biogeochemical model, including its sediment module. Nevertheless, the tendency of a reaction towards glacial physical boundary conditions resulting from the fitting procedure is encouraging and consistent. Ideally, one would need to simulate the glacial ocean circulation in a coupled Earth system model including an ice sheet model over the entire last glacial-interglacial climatic cycle. However, even given such a detailed and realistic velocity would be available, the computing times for carrying out the various sensitivity experiments would be prohibitively large due to the required short time step in such simulations. Nevertheless, in future studies it would be desirable to include also quick alterations of the ocean velocity field, especially changes in ocean overturning. Such short-term climatic changes (time scale of few hundred to thousand years) have been inferred from ice core as well as sediment core analysis known as Dansgaard Oeschger events (Dansgaard et al. (1993), Anklin et al. (1993)) where the coldest events are also marked by large amounts of ice rafted debris in sediment cores (Heinrich events (Bond et al. (1993)). Non-linear ocean-atmosphere dynamics (Barker et al. (2015); Olsen et al. (2005)) would need to be included in respective simulation attempts. Also the representation of sea water temperature changes can be improved. The LGM sea surface temperatures have been accounted for through the respective glacial forcing field underlying the simulation (CLIMAP Project Members (1976); CLIMAP Project Members (1981)). The simulated deep water temperature drop below 1500 m was around 1.2°C on the average (Winguth et al. (1999)) as compared to the pre-industrial/interglacial simulation, with some areas where the temperature difference was up to -2°C, especially in the North Atlantic deep water. Reconstructions of bottom water temperatures through oxygen isotope pore water analysis revealed a temperature decrease of around 2°C at the Carnegie Ridge (Pacific) and the Ceara Rise (Atlantic) (Cutler et al. (2003)) and close to deep water productions sites cooling of deep waters in North Atlantic, South Pacific, and Southern Ocean by about 4-5°C, 2.5°C, and 1.5°C (Adkins et al. (2002)). Consistent with this, bottom water interglacial-glacial temperature changes have been inferred from Mg/Ca paleo-thermometry (Dwyer et al. (1995), Skinner et al. (2003), Roberts et al. (2016)). The modelled sea water temperatures may thus be somewhat higher than the observed ones, especially for the Southern Ocean. It should be note that the circulation change experiment with the biogeochemical model was carried out with preindustrial temperatures (for the biogeochemistry only) in order to separate the temperature and circulation effects properly (and to avoid linear parameter dependencies in the inverse approach). The most difficult aspects to interpret in the results are the reactions of the ecosystem parameters. The small changes in POC and BSi degradation or dissolution rates may be realistic in view of the overall still

modest change in seawater temperatures. The almost vanishing rain ration change in the rank 6 solution and the suggested rain ratio decrease (corresponding to $CaCO_3$ export production decrease) cannot be easily explained. The lack of better information on $CaCO_3$ production during glacial times does not allow us to give any clear answer concerning potential ocean acidification impacts. The reaction of the system during glacial times remains diffuse in our analysis (see also below for the resolution of parameters analysis).

For some $CaCO_3$ tracer records a remarkable improvement in fit resulted when increased dust deposition and the related dilution effect where employed (not shown). However, on average the inverse approach does not suggest a strong effect of the dust deposition changes for providing a better reproduction of the paleo-climate tracers here. This can possibly be due to the simple modelling approach which we take here and to the regional variations in dust deposition which so far cannot be resolved by the input fields.

The increase in carbon overconsumption (change in stoichiometry of carbon to nutrient elements) as suggested in our study confirms earlier results (Heinze and Hasselmann (1993)). This does not necessarily mean that such a change in carbon to nutrient ratio is realistic, for two reasons: the ecosystem changes needed to achieve such a shift would have to be dramatic and the glacial-interglacial atmospheric $CO_2$ change corresponds in amplitude to about oncrease from pre-industrial to present. So far, no respective carbon underconsumption due to the anthropogenic $CO_2$ release has been identified. Rather, the opposite has been suggested for further dramatic increases in atmospheric and sea surface $CO_2$ (Riebesell et al. (2007)). The increase of carbon to nutrient ratios under cold and low-$CO_2$ conditions would contradict the evidence from mesocosm experiments in a Norwegian coastal setting (Riebesell et al. (2007)). The experimental results from the mesocosm studies may potentially be influenced by the specific experiment setting and thus may not be valid at other locations. Earlier, Heinze and Hasselmann (1993) could not well separate the effect of the stimulation of POC export production by a change in stoichiometry or by an increase of the oceanic inventory of dissolved phosphate. Thus, the requirement of a carbon overconsumption in this study may indicate that the biological organic carbon production may have been stimulated by other processes such as fertilisation of ocean productivity by dust-derived micro-nutrients.

We also look at the impact of the rank reduction on the parameter change quantification and the goodness of fit to the observations. The reduction of the rank of our linear system means that not all unknown parameter changes can be determined in an independent way. The resolution matrix of the parameters (cf. eq. 17) identifies which free parameters can still be uniquely determined after the rank reduction and which ones can only be given as linear combinations of each other.

Instead of giving the matrices in terms of numerical values, we visualised the matrices through circles in Figure 9, for the solution with rank 8 (full rank), rank 7, and rank 6. The full rank resolution diagram shows a perfect diagonal with all parameters formally perfectly resolved. For the rank reduction, the resolution for the rain ratio change, and also the BSi dissolution rate change deteriorates strongly. This shows that the linear system is in fact poorly constrained, and that very little can be firmly stated about the glacial-interglacial rain ratio change. Unfortunately, implications for future ocean acidification impacts cannot be deduced from our analysis. This is important information, however. It may be that the quite comprehensive sediment record collection which we employed here, may still be inadequate to address the ocean acidification problem and possibly also the atmospheric glacial $CO_2$ draw-down problem. Parts of the real system may not be represented - be it in the observations, in the

model, or both. Further sedimentary tracer types may be needed in order to make the data base more complete and to resolve all (and possibly more) carbon cycle parameters properly. The trade-off between realistic parameter estimates and goodness of fit becomes obvious when one checks, e.g., the overall estimated reproduction of the atmospheric Vostok ice core curve for the full rank and rank deficient solutions (Figure 10). While the full rank solution gives a quite striking fit to the atmospheric

$CO_2$ curve, already in the rank 7 solution there is a strong decrease in fit, while finally the rank 6 solution (and thus also all solutions with smaller ranks) reveals an almost complete deterioration of the fit. This provides a clear dilemma. In general, the improvement of the fit for a simultaneous optimisation of all parameters is quite weak except for atmospheric $pCO_2$ (see summary in Table 6). However, our 3-D biogeochemical ocean model is not detailed enough to capture all aspects of the glacial ocean correctly. For example, our representation of the circulation variations is simplified. Also it may be possible that

we have missed one or more key processes, which may have caused or contributed to the simultaneous changes in the paleo-climate tracer distributions. Further, we did not carry out any regional differentiation concerning the perturbation of governing parameters but used global mean changes. More detailed data assimilation schemes and the use of higher resolution coupled biogeochemical-physical models will be an option for the future for better reproduction of the sediment core data. The price for this will be the many more degrees of freedom that need to be constrained correctly. It remains a possibility that the glacial

biological $CaCO_3$ shell material production was indeed considerably smaller than the interglacial production, such that this parameter change did not cause any major imprint on the paleo-climate tracer combination used in this study. In addition, also some of the sedimentary records may reflect specific local conditions which cannot be spatially resolved by our coarse model. Such special local conditions may include sediment focusing due to specific bathymetric features, smaller scale dynamic flow conditions such as localised upwellings, and meandering frontal systems.

We also analysed the change in simulated tracer records as compared to the control run (Table 7). As one can see, in spite of the poor fit (except for the atmospheric $pCO_2$ record), deviations from the tracer output records of the model control run occurred. These tend to get smaller with the reduction of the rank. Correlation coefficients between observations and predicted tracer values from the linear response model are given in Figure 11. While for a series of single records the correlation is good, for other single records the correlation is poor or even anticorrelations resulted. This is especially the case for the planktonic

$\delta^{13}C_{planktonic}$ records. This deficiency can be in part explained through the coarse model resolution. If the regional extent of upwelling zones is different between model and reality also the respective surface tracers for paleo-productivity as recorded in the sediment cores will show respective differences.

        In previous studies on the glacial-interglacial changes in the ocean carbon cycle, often hypotheses involving one specific mechanism were presented or the multitude of potential mechanisms was reviewed. Studies where the simultaneous contri-

butions from several processes to glacial carbon dynamics have been discussed are relatively scarce. Brovkin et al. (2007) employed an Earth system model of intermediate complexity including oceanic and terrestrial biogeochemical modules to test the impact of simultaneous changes on the atmospheric $CO_2$ concentration. Their results are fairly consistent with those of this study. According to Brovkin et al. (2007), largest contributions to the $CO_2$ drawdown came from circulation and SST changes as well as from a strengthening of the biological pump through improved nutrient utilisation, while the land outgassing

amounted to an atmospheric $pCO_2$ increase by 15 ppm. Rain ratio changes contributed to about 15 ppm, a process also cited

as a less certain mechanism by Brovkin et al. (2007). In addition, they report secondary changes of atmospheric $pCO_2$ due to weathering, sea level change, and changes in sedimentation (shallow water vs. deep water).

Recent studies focused again on the mineral dust hypothesis involving increased iron supply especially to the Southern Ocean (originally revived by Martin et al. (1994) and Berger and Wefer (1991)) and a respective regional strengthening of biological production and carbon export. Increased LGM aerosol iron flux to the Southern Ocean could be corroborated by Conway et al. (2015). With our approach, in a separate experiment (not shown in this study where we only consider dust for solution of sedimentary material) we also had tested increased ocean productivity and related surface nutrient drawdown due to changing dust deposition. The inverse approach did not favour this process. This is fairly consistent with the modelling study by Lambert et al. (2015) arriving at a direct effect of the iron induced biological pump strengthening of less than 10 ppm and delayed effect due to carbon burial and carbonate compensation by about 10 ppm.

Inspired by the suggestion of temperature-dependent export production of Laws et al. (2000), Matsumoto et al. (2007) quantify in a further single mechanism study the effect of temperature dependent remineralisation on the atmospheric $CO_2$ using an ocean biogeochemical model. According to their findings, an LGM atmospheric $CO_2$ decline by 30 ppm would be possible through this process. In our study, the parameter change of the POC remineralisation rate with temperature forcing did not result in a similarly likely drawdown when tested in the inverse approach against evidence from the sedimentary record. Rather our work suggests that simple temperature and $pCO_2$ dependent changes of ocean physics as well as biogeochemistry do not straightforwardly translate into atmospheric $CO_2$ changes and respective sedimentary imprints, but that the problem is more complicated.

The Southern Ocean has been established to be one of the key regions for regulating glacial-interglacial carbon dynamics. Apart from processes involving dissolved iron and nutrients, especially the physical dynamical processes - and hence stratification, deep water production, upwelling, water mass formation, and lateral advection - have been considered in conjunction with the physical/chemical and biological carbon pumps. Special emphasis has been placed on a northward shift of the westerlies wind forcing at the LGM as compared to today (Toggweiler et al. (2006); Watson and Garabato (2006); Watson et al. (2015)). The general idea is that a northward shift of Southern Ocean upwelling leads to reduced $CO_2$ outgassing and enhanced carbon export to the deep waters resulting in a deep ocean accumulation of organic matter from the surface and hence a vertical "fractionation" of carbon as well as nutrients as described already by Boyle (1988a) and Boyle (1988b). This general view is corroborated also by recent proxy data findings (for the Southern Ocean by Gottschalk et al. (2016); also for the North Atlantic by Hoogakker et al. (2015)). Refined Southern Ocean dynamics could also improve the results of our studies, but we have been limited to the flow fields available. In general, Southern Ocean flow field and tracer simulations show traditionally a large intermodel spread (Broecker et al. (1998); Roy et al. (2011); Orr (2002)). One of the reasons is the complex interplay of sea-ice as well as wind forcing and also the subgrid-scale parameterisation of convection events which occur on narrow regional scales in reality (Gordon (1978)). Still, simulating the Southern Ocean flow field and mixing remains a key challenge even for the modern ocean (Farneti et al. (2015): Downes et al. (2015); Mignot et al. (2013); Abernathey et al. (2016)).

## 6 Conclusions

In our study we combine a comprehensive sedimentary data base from the paleo-climatic sediment core record with a coarse resolution BOGCM. We assume that the governing carbon cycle parameter variations over the past climatic cycle follow the same temporal pattern. As the paleo-record of atmospheric $pCO_2$ and surface temperature show a strong correlation (Siegenthaler et al. (2005)), we cannot decide in general whether the parameter variations are caused by physical forcing (temperature), chemical forcing ($CO_2$), or both simultaneously. With such approximation, we can reduce the search for unknown parameter changes to the maximum amplitude of these changes at the LGM. By employing 79 observed paleo-climatic tracer data curves (with altogether 20,540 formal data points) for determination of 8 unknown maximum parameter excursions with respect to the preindustrial situation, we arrive at a formally completely overdetermined linear response model for estimating the parameter changes. We can confirm the quantification for mean global maximum SST change (-5 K) and carbon loss from terrestrial systems (-430 PgC) to the atmosphere as suggested by earlier studies. Further, our model-data combination clearly identifies a substantial increase in the biological organic carbon export during glacial times, though the underlying process may not be a carbon to nutrient overconsumption. The response of glacial-interglacial changes in the biological $CaCO_3$ production remain uncertain. $CaCO_3$:$C_{org}$ rain ratio reductions may have contributed to the glacial $CO_2$ reduction in the atmosphere, but such rain ratio changes cannot be resolved with our linear response model. This may be due to model deficiencies in our approach, or due to lack of information from the present data base for this purpose. With our combination of model results and observations no overall answer concerning the mechanisms behind the glacial $pCO_2$ drawdown can be provided. Simple synchronous temperature dependent or atmospheric $pCO_2$ dependent forcing fields and governing parameter time series of the same temporal pattern appear to not be able to reproduce the marine sediment core record satisfactorily.

**Data sets:** The observational sediment core data sets have been taken from the literature. For the comprehensive compilation of $CaCO_3$ sediment core data, please, contact B. Hoogakker.

## Appendix A: Model details

Major process parameterisations of the biogeochemical model used in this study are summarised below. The model is the annually averaged version of the HAMOCC model (Maier-Reimer (1993); Maier-Reimer et al. (2005)) and has been described elsewhere (Heinze and Maier-Reimer (1999); Heinze et al. (1999); Heinze et al. (2003); Heinze et al. (2009)). Several free model parameters have been tuned anew for this study as compared to earlier studies. A number of important parameters are summarised in Table A1.

**Air-sea gas exchange**

The atmospheric model reservoir and the ocean surface layer exchange both $CO_2$ and oxygen. The flux of $CO_2$ across the air-sea interface is simulated as follows:

$$F_{CO2} = k_{CO2} \cdot (pCO_{2,air} - pCO_{2,water}) \tag{A1}$$

where $F_{CO2}$ is the carbon dioxide flux across air/sea interface and $k_{CO2}$ is the specific gas exchange rate. The $CO_2$ partial pressure is calculated from the free carbon dioxide concentration (sum of aqueous $CO_2$ and carbonic acid) in seawater with Henry's law through use of the solubility $\alpha$ (Weiss (1974)):

$$pCO_{2,water} = \alpha \cdot [CO_2] \tag{A2}$$

For the oxygen flux the following approach is pursued:

$$F_{O2} = k_{O2} \cdot (C_{O2,equilibrium} - C_{O2,actual}) \tag{A3}$$

$$C_{O2,equilibrium} = f(T,S) \cdot C_{O2,atmosphere}(t)$$

where $F_{O2}$ is the net gas flux between sea surface and atmosphere, $k_{O2}$ is the mean gas transfer velocity, $C_{O2,equilibrium}$ and $C_{O2,actual}$ are the oceanic oxygen concentrations for solubility equilibrium with the atmosphere and the actual observed (or modelled) value, $f(T,S)$ is the measured solubility as a function of sea water temperature and salinity, and $C_{O2,atmosphere}(t)$ is the tropospheric $O_2$ concentration, where $f(T,S)$ is taken from Weiss (1970). $O_2$ gas exchange is a result of the analytical solution of the differential equation

$$\frac{dC}{dt} = F/\Delta z,$$

where $C$ is the gas concentration in [mole/cm$^3$], F the flux in [mole(cm$^2 \cdot$ s)] of gas into (or out of) a control volume of 1 cm length, 1cm width and thickness $\Delta z$ in [cm]. The numerical formulation is

$$C^{new} = C_{equilibrium} \tag{A4}$$
$$+ (C^{old} - C_{equilibrium}) \cdot e^{(-k_{av}/\Delta z) \cdot \Delta t}$$

where $\Delta t$ is the time step (here 1 year) and $\Delta z$ is the thickness of the layer affected by gas exchange ($\Delta z$ is set equal to 50 m in cases of hydrostatic stability and to the maximum depth of the convective layer else). The model atmosphere is represented through a 1-layer box over each grid point. At every time step, zonal averages are determined for the atmospheric concentrations. Gas transport is simulated through meridional diffusion only (as in reality the intrahemispheric tropospheric mixing time is much shorter than the time step of one year applied here).

**Biogenic particle export production and particle flux through the water column**

As a consequence of the annual averaging, the model simulates only export production of biogenic particles. Particle production is assumed to exclusively take place in the model surface layer representing the euphotic zone. Phosphate serves as the ultimate biolimiting nutrient. The nitrogen cycle is not simulated explicitly. POC (particulate organic carbon) and opal export

production rates are predicted using Michaelis Menten kinetics for nutrient uptake (e.g., Parsons and Takahashi (1973)):

$$P_{POC} = \frac{V_{max}^{POC} \cdot [PO_4^{3-}]^2 \cdot Red(C:P)}{K_s^{POC} + [PO_4^{3-}]} \tag{A5}$$

$$P_{opal} = \frac{V_{max}^{opal} \cdot [Si(OH)_4]^2}{K_s^{opal} + [Si(OH)_4]} \tag{A6}$$

where $P_{POC}$ and $P_{opal}$ are the POC and opal export production rates [molel$^{-1}$ yr$^{-1}$], $Red(C:P)$ is the Redfield ratio C:P, $V_{max}^{POC}$ and $V_{max}^{opal}$ are the maximum uptake rate of phosphate and silicic acid from the water column [yr$^{-1}$], and $K_s^{POC}$ as well as $K_s^{opal}$ are the respective half saturation constants. The parameters $V_{max}^{POC}$, $V_{max}^{opal}$, $K_s^{POC}$, and $K_s^{opal}$ are prescribed as a function of sea surface temperature following Heinze et al. (2003).

CaCO$_3$ export production depends on the local production ratio $P_{opal}/P_{POC}$. CaCO$_3$ export starts to increase gradually when $P_{opal}/P_{POC}$ sinks below a threshold value $S_{opal}$, i.e., when the silicic acid concentration is becoming too small to support diatom growth exclusively:

$$P_{CaCO_3} = P_{POC} \cdot R \cdot (1 - \frac{P_{opal}/P_{POC}}{S_{opal}}), \tag{A7}$$
$$for \quad P_{opal}/P_{POC} < S_{opal};$$

$$P_{CaCO_3} = 0, \tag{A8}$$
$$for \quad P_{opal}/P_{POC} \geq S_{opal}$$

where $R$ is the maximum possible rain ratio C(CaCO$_3$):C(POC) and $S_{opal}$ is the threshold value of $P_{opal}/P_{POC}$ for gradual onset of CaCO$_3$ production.

Particle fluxes and particle degradation are simulated through mass balance equations for sinking particulate matter $M_{settle}$, where $M_{settle}$ stands for the different particle species $POC_{settle}$, $CaCO3_{settle}$, $opal_{settle}$, and $clay_{settle}$ respectively. The general formulation for the mass balance $M_{settle}$ becomes then:

$$\frac{d\,M_{settle}}{dt} = gain - loss \tag{A9}$$

$$\frac{d\,M_{settle}}{dt} = P_M - \frac{w_M}{\Delta z_0} \cdot M_{settle} - r_M \cdot M_{settle}$$
$$(for\ surface\ layer)$$

$$\frac{d\,M_{settle}}{dt} = w_M \cdot \frac{\partial M_{settle}}{\partial z} - r_M \cdot M_{settle}$$
$$(for\ other\ layers)$$

where $\Delta z_0$ is the thickness of the euphotic zone [m], $w_M$ is the particle settling velocity [m yr$^{-1}$], $r_M$ is the reaction rate constant [yr$^{-1}$] for degradation of particulate matter, and $P_M$ is the export production rate in the top layer, where again "$M$" denotes one of the particle species POC, CaCO$_3$, opal, and clay respectively. $P_{clay}$ is the dust input from the atmosphere which is prescribed for the control run according to the modern dust deposition field from (Mahowald et al. (1999)). Clay material is considered as chemically inert. The system of equations (A9) is solved through an implicit numerical scheme. CaCO$_3$ dissolution in the water column is performed depending on the prevailing carbonate saturation using the following rate

constant:

$$r_{CaCO3} = k_{CaCO3} \cdot (1 - \Omega) \tag{A10}$$

where $k_{CaCO3}$ is a fixed standard rate constant, and $(1 - \Omega) = ([CO_3^{2-}]_{sat} - [CO_3^{2-}])/[CO_3^{2-}]_{sat}$ is the degree of undersaturation. Similar to Heinze et al. (2009) we apply a minimum value of 0.062 for the undersaturation, thus allowing for some
CaCO$_3$ dissolution in oversaturated waters (as a simplification, our model simulates only calcite and not the meta-stable form aragonite or high Mg calcites). Opal dissolution is formulated to depend on the deviation from the saturation concentration for opal in seawater (following Ragueneau et al. (2000); where, in the formulation below, the effect of changes in free surface area of reactive opal is formally lumped into $k_{opal}$ here):

$$r_{opal} = k_{opal} \cdot (Si_{saturation} - Si_{actual})/(1 \quad mole \quad l^{-1}) \tag{A11}$$

with $k_{opal}$ a standard upper limit opal degradation rate, $Si_{saturation}$ the water column saturation value for silicic acid, and $Si_{actual}$ the actual silicic acid value at a model grid point (the division by $1 mole l^{-1}$ is needed in order to match the units correctly).

The losses in particulate matter according to eq. (A9) for settling particulate material are mirrored by respective source terms for the dissolved species within the water column (for TAlk, DIC, phosphate, oxygen, and silicic acid). The remainder
of particles which are not subject to degradation within the water column are deposited onto the top sediment layer (see sedimentary source term $Q$ further below).

**Sediment pore water chemistry and diffusion**
Early diagenetic processes are simulated as described in Heinze et al. (2009), following the efficient numerical formulations of Maier-Reimer et al. (2005) for the implicit simultaneous treatment of pore water transport and pore water reactions as well as the vertical sediment advection. The porosity depth profile is fixed using numerical values of Ullman and Aller (1982). The basic sediment model concept is expressed through the equilibrium expressed by the following equation:

$$sediment \; accumulation = deposition - redissolution$$

The mass balance equations for a solid sediment component (expressed in moles per unit volume of sediment $S_*$) and the related concentration of the corresponding dissolved substance $C$ within the pore water are:

$$\frac{dS_*}{dt} = D_B \frac{\partial^2 S_*}{\partial z^2} - \frac{\partial}{\partial z}(w \cdot S_*) - G \tag{A12}$$

$$\frac{dC}{dt} = \frac{\partial}{\partial z}(D_W \frac{\partial C}{\partial z}) + G \tag{A13}$$

where $D_B$ is the diffusion coefficient for bioturbation, $w$ the vertical advection velocity of solid compound, and $G$ the reaction
rate (with $S_*$, $C$, and $G$ reported here in relation to full sediment volume of a given sediment layer for sake of simplicity; within the model, the varying volumes for dissolved and solid substances following the porosity profile are taken into account).

The prognostic equations for solid sediment concentrations, $S_*$ (organic carbon, $CaCO_3$, opal, and clay), and for dissolved components $C$ (TAlk, DIC, phosphate, oxygen, and silicic acid) are coupled through the respective reaction rates. The diffusive transport of pore water tracer concentrations is simulated via the respective deviations from saturation in parallel with the reduction of this deviation due to chemical pore water reactions:

$$5 \quad \frac{dU}{dt} \quad = \quad \frac{\partial}{\partial z}(D_W \frac{\partial U}{\partial z}) - G \tag{A14}$$

where $U$ is the deviation of the saturation concentration, $C_{sat}$, from the actual concentration $C$) [mole $l^{-1}$], $G$ is the reaction rate (sink for $U$ from dissolution of solid material) [mole $l^{-1}$ $yr^{-1}$] and $D_W$ is the diffusion coefficient. As diffusion coefficient, $D_W$, we employ a general value of $8 \cdot 10^{-6}$ cm$^2$/s for all pore water tracers (this value is in the range of the values given for various pore water species in the work of Li and Gregory (1974). The solid sediment components change due to pore water reactions and particle deposition corresponding to eq. A14:

$$\frac{dS_*}{dt} \quad = \quad -G + Q \tag{A15}$$

where $S_*$ is the solid sediment component expressed in the same units as $U$ [mole $l^{-1}$], $G$ is the reaction rate (sink due to dissolution) [mole $l^{-1}$ $yr^{-1}$], and $Q$ is the deposition flux from particle rain [mole $l^{-1}$ $yr^{-1}$] (the latter occurs only in the top sediment layer). The amount $G$ of solid matter which can be dissolved per unit of time is a function of parameter $r_c^*$, the deviation from saturation concentration $U$, and the amount of solid material available $S_*$:

$$G = r_c^* \cdot U \cdot S_* \tag{A16}$$

where for $r_c^*$ we have:

$$r_c^* = \frac{r_c}{C_{sat}} \tag{A17}$$

with $r_c$ being the reaction rate constant [$yr^{-1}$] and $C_{sat}$ the saturation concentration in solution [mole $l^{-1}$]. For opal dissolution a solubility $C_{sat}$ of 800 $\mu$mole $l^{-1}$ is used [cf. Dixit et al. (2001), and for organic carbon a constant formal value of 100 $\mu$mole $l^{-1}$ is used (details of anaerobic respiration and respective effects such as denitrification and sulphate reduction are not taken explicitly into account). For the sedimentary $CaCO_3$, the $CO_3^{2-}$ saturation concentration is now also computed through the revised inorganic carbon system quantification following Dickson et al. (2007) (see eq. A10). The discretised versions of the coupled equations (A14 and A15) are solved implicitly through a numerical back-substitution formulation. Transport of dissolved tracers through pore water diffusion is not carried out as a boundary condition problem. Rather we formally include the lowermost wet grid box in the water column directly over the model seafloor in the pore water diffusion scheme. Thus the free water column and the sediment pore waters directly exchange matter with each other.

**Bioturbation**

Bioturbation is simulated as in Heinze et al. (2009) through vertical "diffusion" of the solid substances. This step is carried out after the vertical advection step described further below. The solid matter compounds are slowly mixed in proportion to the

prevailing weight fractions in the layers overlying each other:

$$\frac{dS_*}{dt} \quad = \quad \frac{\partial}{\partial z}(D_B \frac{\partial S_*}{\partial z}) \tag{A18}$$

with $D_B$ being the bioturbation coefficient. Non-local mixing (e.g., Boudreau and Imboden (1987)) is neglected here.

5 **Advection of solid sediment weight fractions and sediment accumulation** Vertical advection of sediment is performed as in Heinze et al. (2009), where the direction of sediment accumulation depends on the amount of the matter deposited onto the top sediment layer, and the redissolution rate of matter within the sediment mixed layer. The four different weight fractions clay, $CaCO_3$, opal, POC, and POP are shifted downward or upward according to the gain from the deposition flux and the gaps which are created from dissolution. In case of very strong corrosion in the bioturbated zone, only clay is transported 10 upward from below the sediment mixed layer. For an illustration of the mode of sediment advection for the different cases (a) burial $>$ rain $-$ redissolution (accumulation of sediment) and (b) burial $<$ rain $-$ (erosion of sediment), please, see Figure 2 in Heinze et al. (2009). The vertical sediment advection velocity $w$ and also its sign (see eq. A12) depend on the sediment deposition from the water column and the strength of the pore water chemical reactions.

*Acknowledgements.* We would like to thank David Archer and an anonymous reviewer for their expert comments which improved this 15 manuscript. Thanks are due to many colleagues and friends. Ernst Maier-Reimer (Max Planck Institute for Meteorology) provided the basic HAMOCC model and crucial advice on efficient numerical modelling. Marion Gehlen and James Orr provided help for revising the model inorganic chemistry treatment for using the appropriate pH scale. Thanks are due to Natalie Mahowald for providing the dust deposition fields. Elizabeth Farmer's language revision in busy times is highly appreciated. Arne Winguth is supported by grant NSF-OCE 1536630 and NSF-EAR 1636629. This work was partially supported through the University of Bergen (sabbatical funds for C. Heinze), through the 20 "European Project on Ocean Acidification" (EPOCA; which received funding from the European Community's Seventh Framework Programme (FP7/2007-2013) under grant agreement n$^o$ 211384). The Research Council of Norway supported this study through the nationally coordinated project "Earth system modelling of climate variations in the Anthropocene" (EVA; grant no. 229771) as well as project "Overturning circulation and its implications for global carbon cycle in coupled models"(ORGANIC; grant no. 239965). This is a contribution to the Bjerknes Centre for Climate Research (Bergen, Norway). A part of the computations where carried out under project NN2980K at the 25 Norwegian Metacenter for Computational Science NOTUR and its dedicated storage and archiving project NorStore (NS2980k).

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

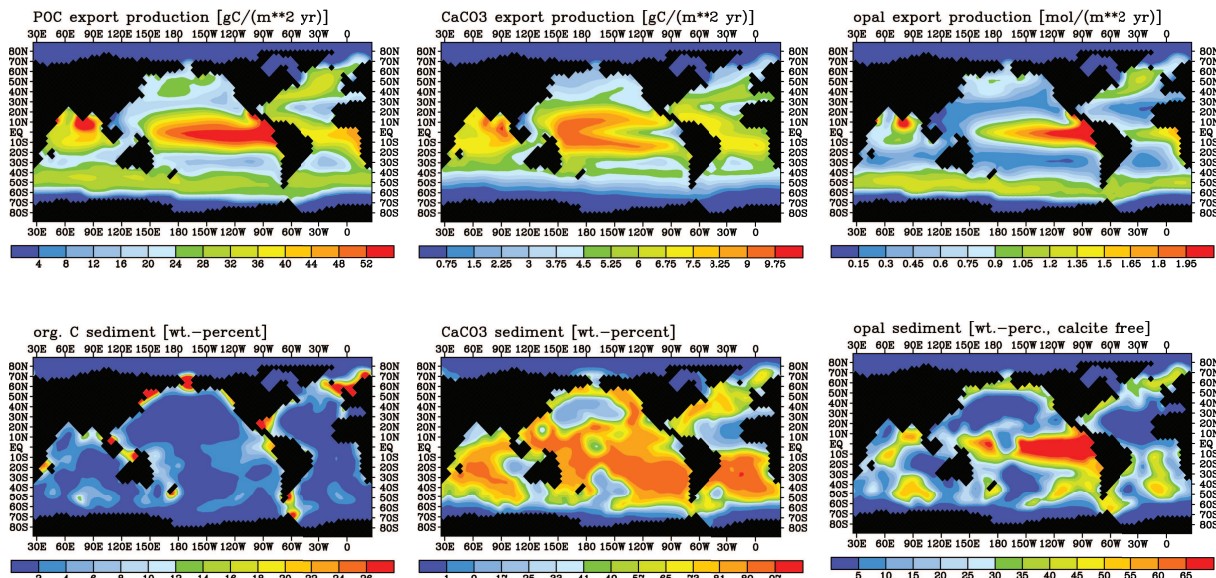

**Figure 1.** Export production rates and average surface sediment distributions for the control run.

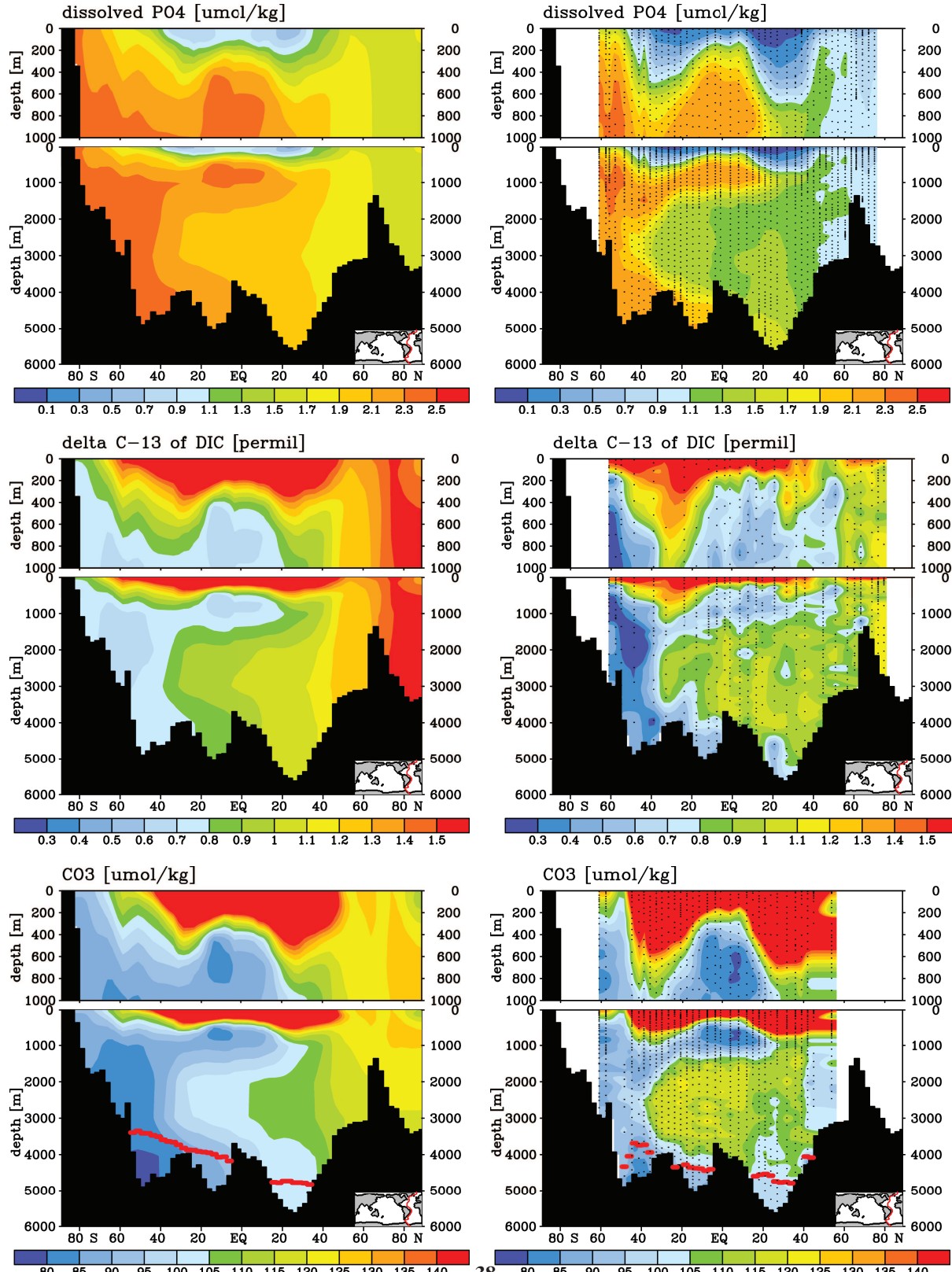

**Figure 2.** Atlantic Ocean cross sections for the model control run (left column) and observations from the GEOSECS programme (right column). Top: Dissolved phosphate. Centre: $\delta^{13}$C of total dissolved inorganic carbon. Bottom: Carbonate ion concentration.

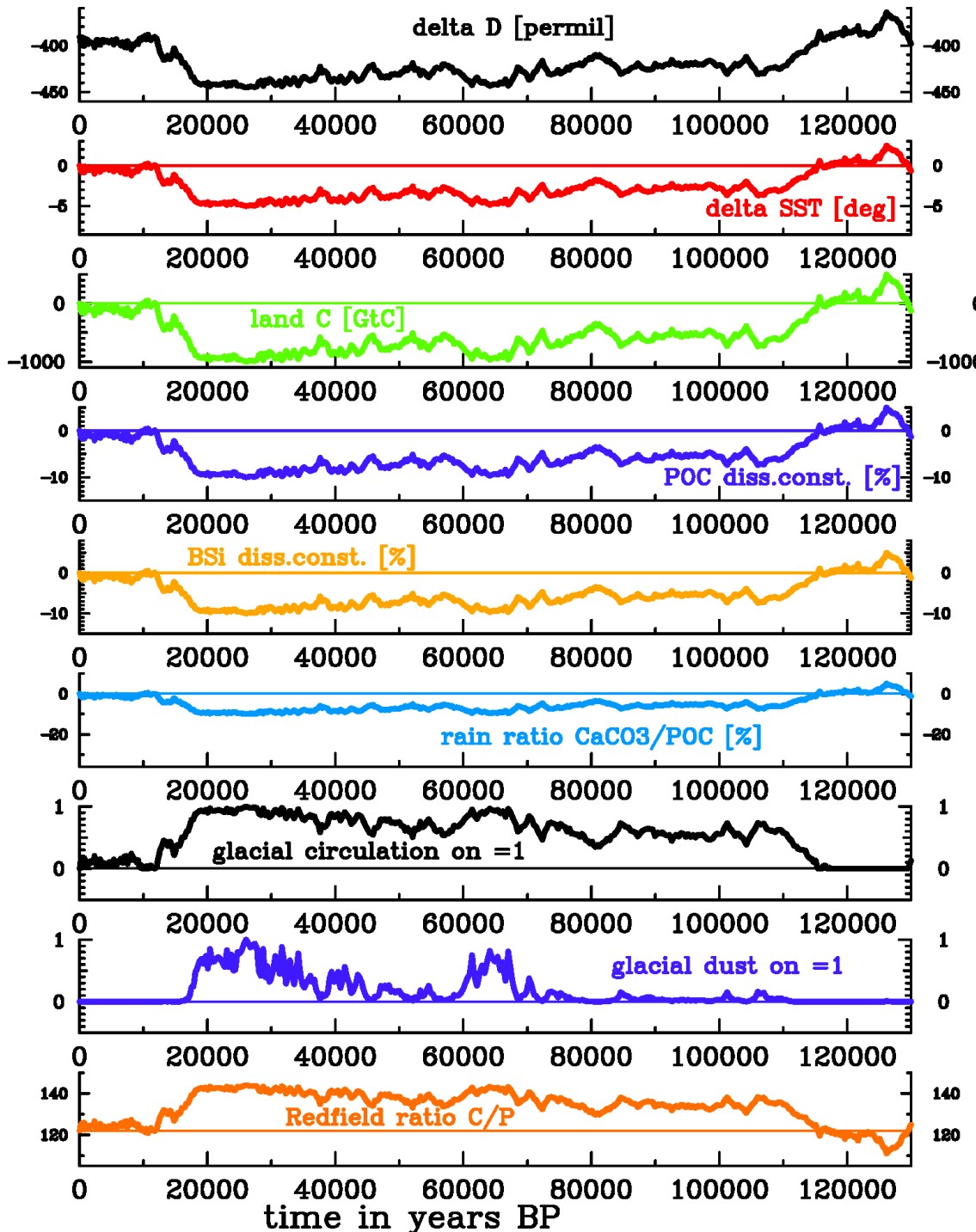

**Figure 3.** Prescribed parameter changes for the sensitivity experiments with the biogeochemcial ocean model.

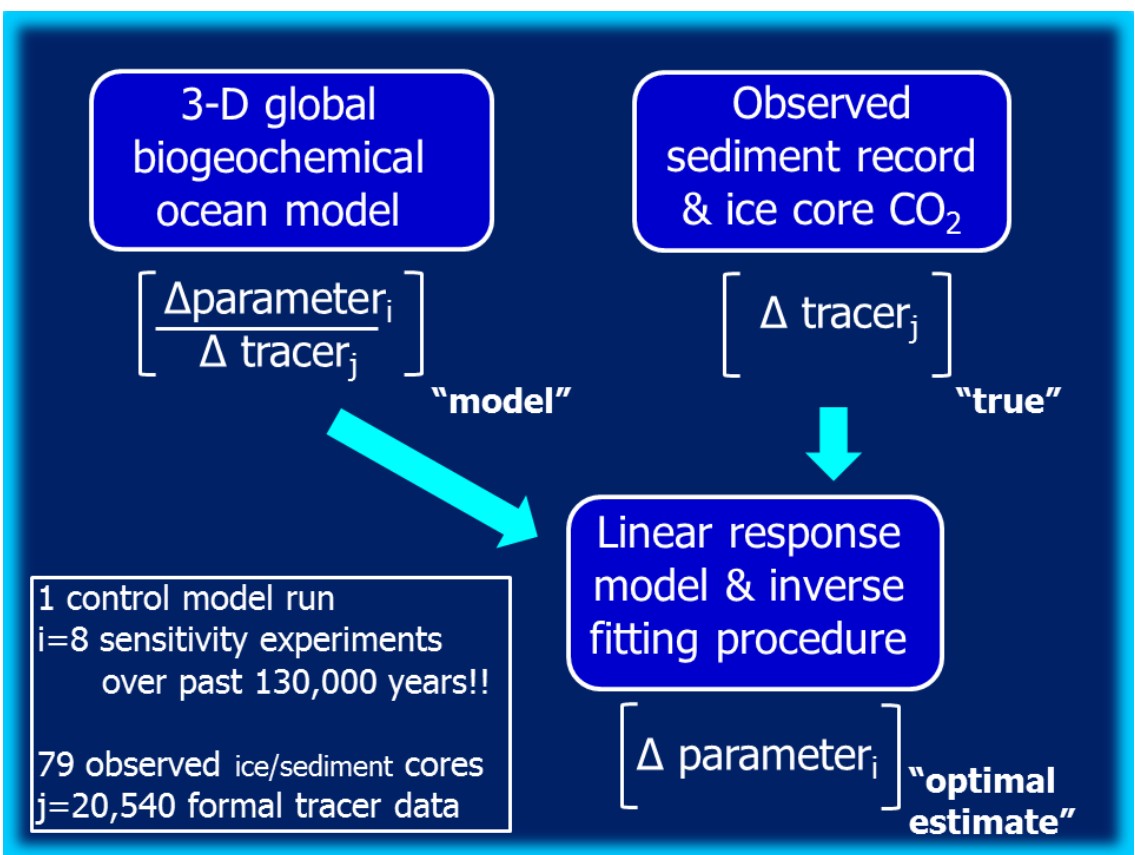

**Figure 4.** Fitting procedure combining the observed paleo-climate data and the model derived information on how the downcore distributions of the various cores look for the different sensitivity experiments.

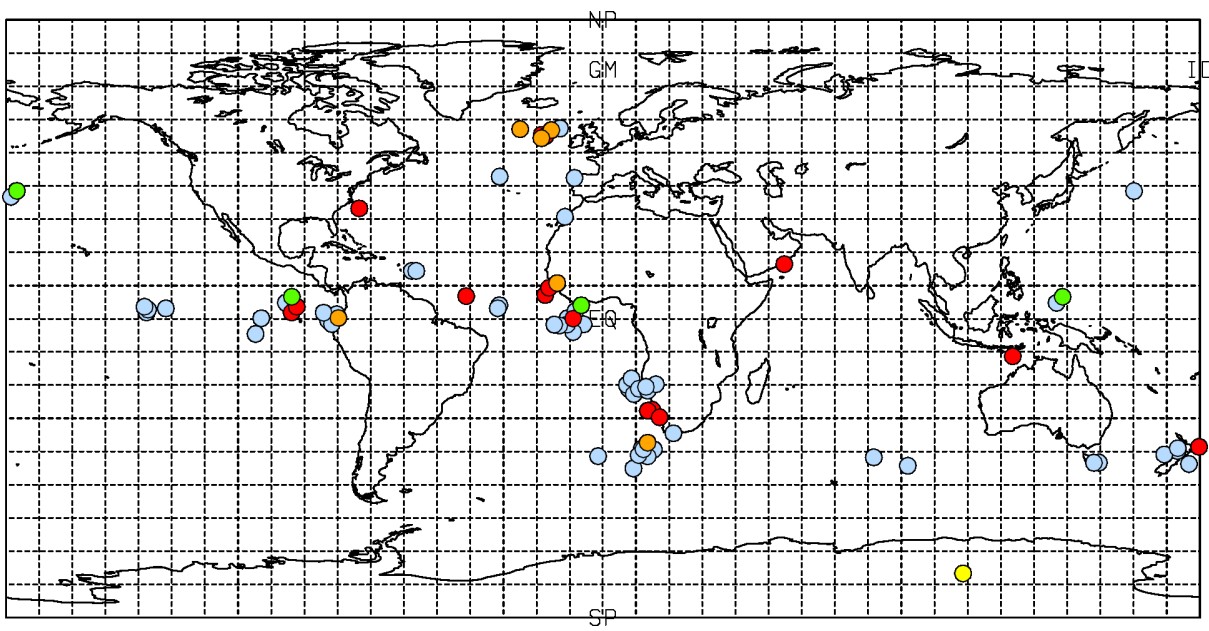

**Figure 5.** Locations of the paleo-climate observations (sediment core, ice core) as used in the inverse approach: atmospheric $CO_2$ (yellow), $\delta^{13}C_{benthic}$ (red), $\delta^{13}C_{planktonic}$ (orange), BSi wt-% (green), and $CaCO_3$ wt-% (blue).

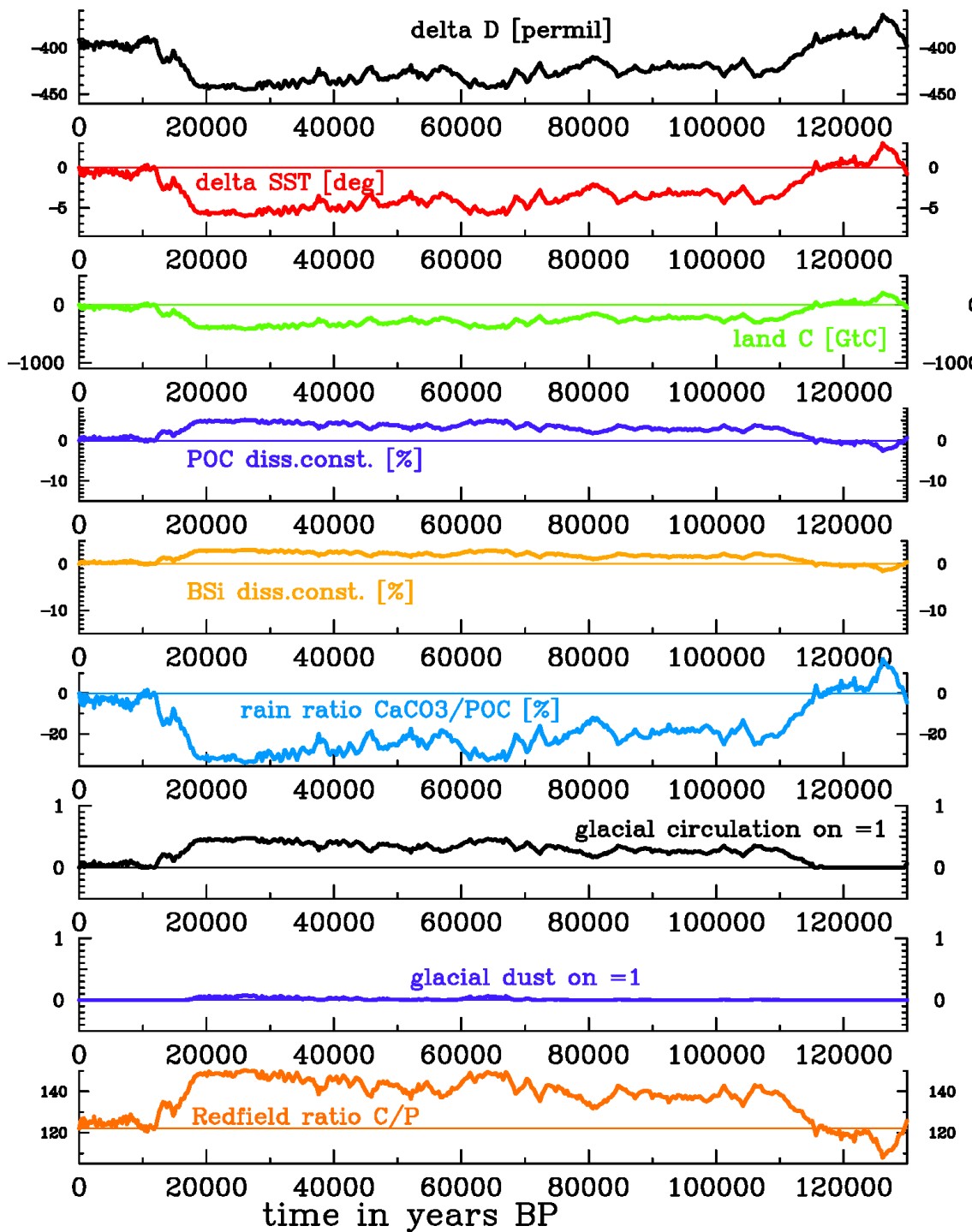

**Figure 6.** Resulting parameter variations of the past climatic cycle for the full rank solution of the linear response model.

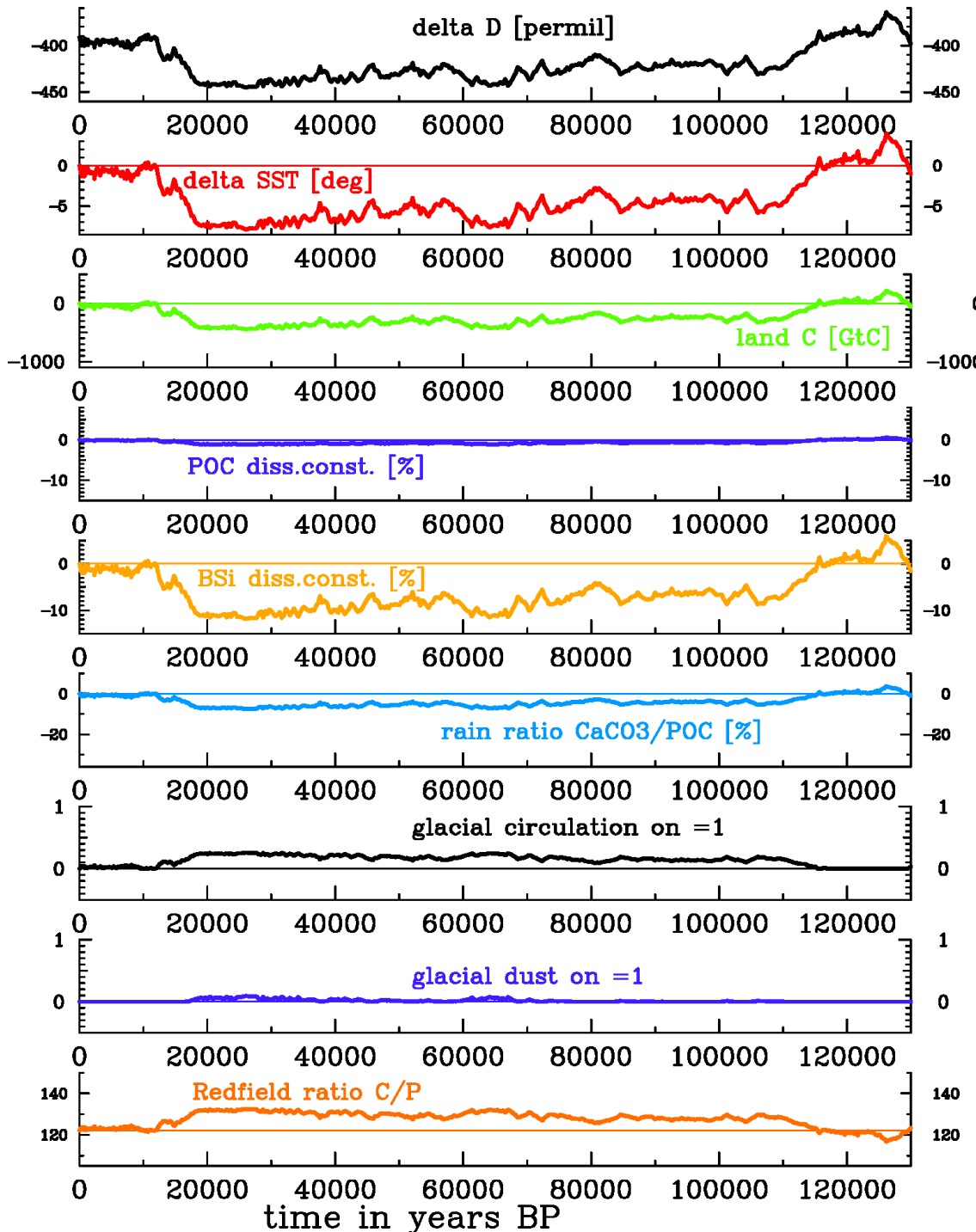

**Figure 7.** Resulting parameter variations of the past climatic cycle for the rank 7 solution of the linear response model.

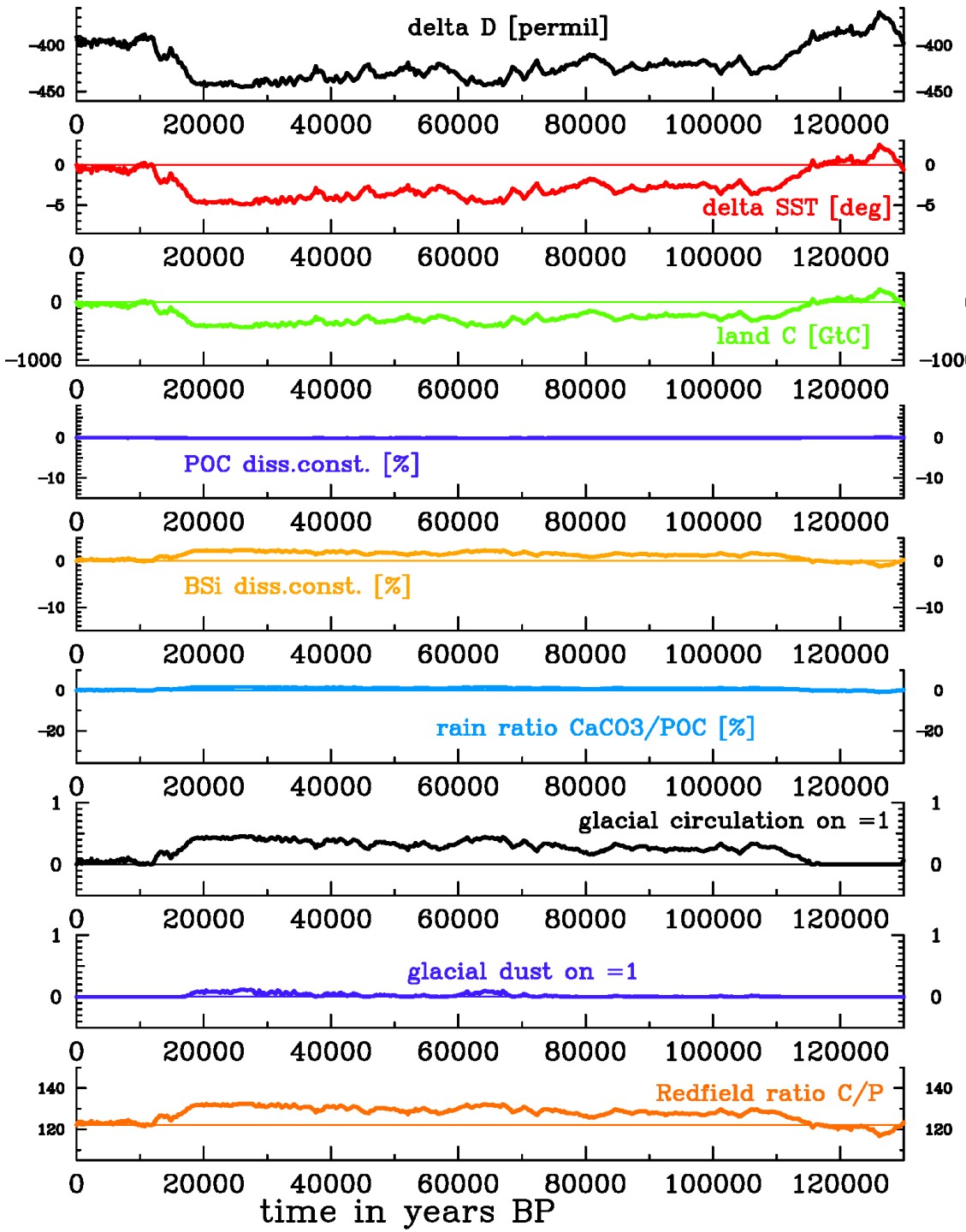

**Figure 8.** Resulting parameter variations of the past climatic cycle for the rank 6 solution of the linear response model.

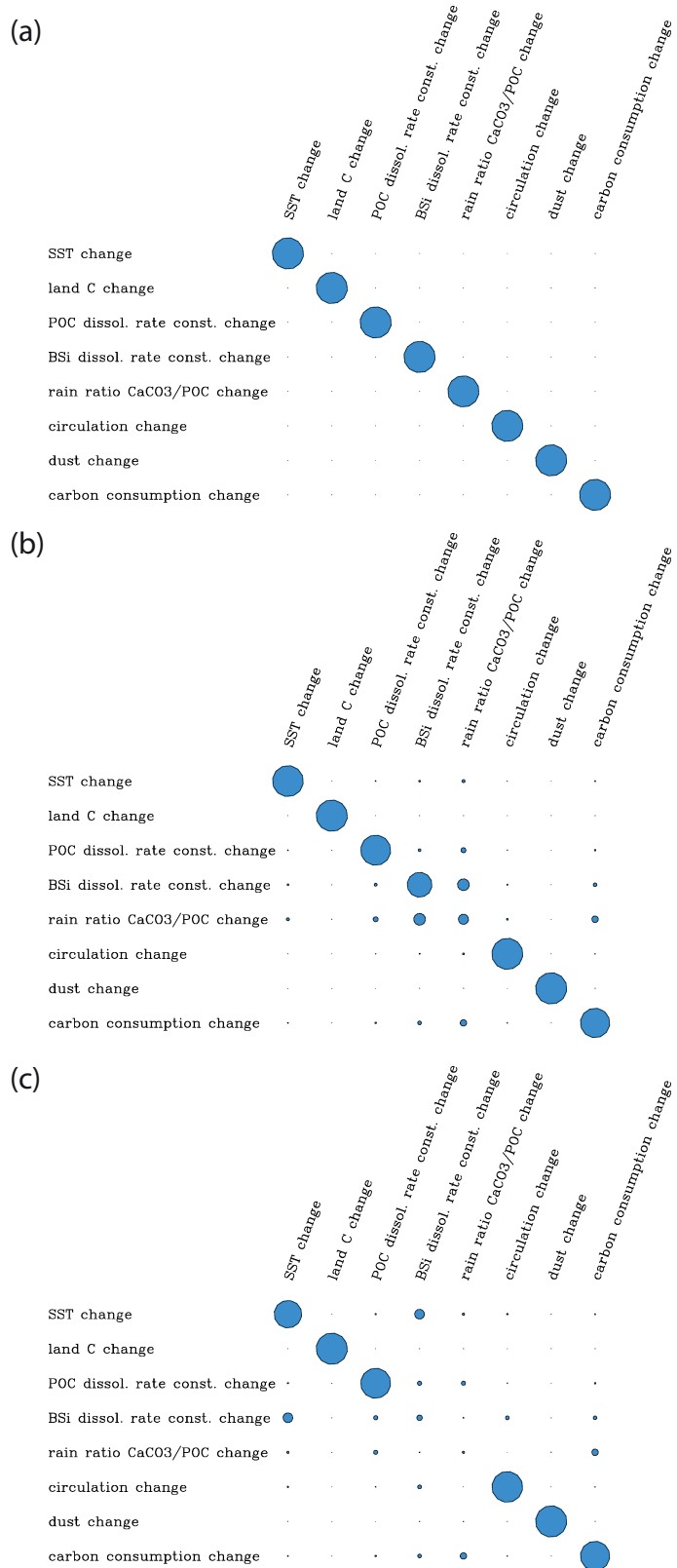

**Figure 9.** Resolution matrices for the full rank (rank 8) (a), rank 7 (b), and rank 6 (c) solutions. The CaCO$_3$:C$_{org}$ rain ratio variations cannot be anymore determined for a realistic length of the solution vector of parameter variations.

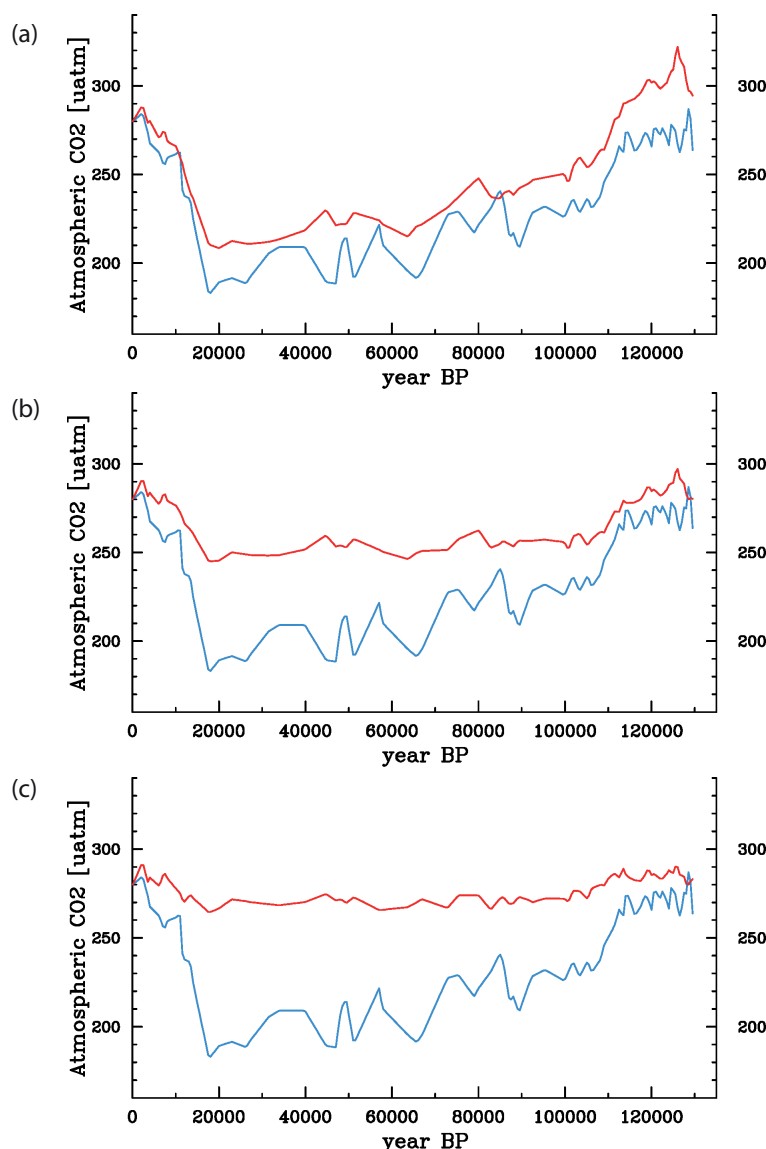

**Figure 10.** Goodness of fit for the full rank (rank 8) (a), rank 7 (b), and rank 6 (c) solutions (blue line: observations; red line: inverse modelling result). The example is for the predicted evolution of atmospheric $CO_2$. The observed record comes from the Vostok ice core, Antarctica (Barnola et al., 1987). The rendering of realistic parameter variations in the rank deficient solutions is accompanied by a strong reduction in goodness of fit.

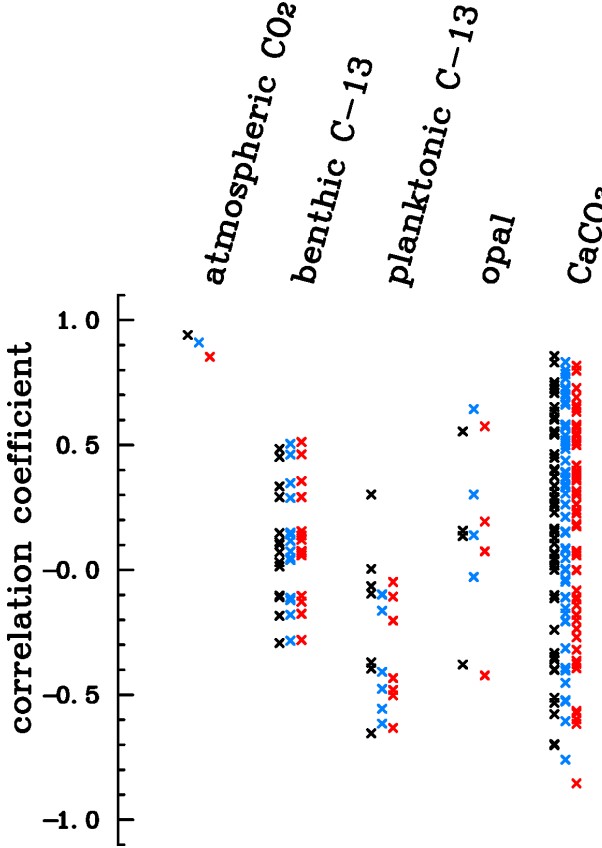

**Figure 11.** Correlation coefficients between observations and predicted tracer values from the linear response model. The correlation coefficients are shown from left to right for the various tracer types. Each record is represented by one small cross. Values for the full rank solution are given in black, values for rank 7 in blue, and values for rank 6 in red.

**Table 1.** Global bulk numbers for the model control run.

| Global Value | Control Run Value |
| --- | --- |
| atmospheric $pCO_2$ [$\mu$atm] | 284.0 |
| POC export production [PgC/yr] | 9.60 |
| POC deposition to sediment [PgC/yr] | 0.088 |
| POC accumulation in sediment [PgC/yr] | 0.060 |
| POC accumulation in sediment [teramole C/yr] | 5.000 |
| $CaCO_3$ export production [PgC/yr] | 1.72 |
| $CaCO_3$ deposition [PgC/yr] | 0.37 |
| $CaCO_3$ accumulation [PgC/yr] | 0.32 |
| $CaCO_3$ accumulation [teramole C/yr] | 27.0 |
| opal export production [teramole Si/yr] | 251 |
| opal deposition [teramole Si/yr] | 13.7 |
| opal accumulation [teramole Si/yr] | 4.5 |

**Table 2.** Parameter changes $x_j$, $j$=1, …, $n$; $n$=8 as imposed in the sensitivity experiments with the full 3-D biogeochemical model relative to the control run parameter choice. The $x_j$ are the maximum amplitude of change for the respective strongest excursion of the observed $\delta$D curve at the LGM.

| j | parameter description | value of param. change $x_j$ |
|---|---|---|
| 1 | sea surface temperature for computation of the chemical and biological constants: | -5° |
| 2 | loss in carbon stock of the terrestrial biosphere: | -1000 PgC |
| 3 | dissolution rate constants of POC: | -10% |
| 4 | dissolution rate constants of BSi: | -10% |
| 5 | export production rain ratio $CaCO_3$:POC: | -10% |
| 6 | 3-D oceanic velocity field: | 100% glacial field |
| 7 | glacial dust deposition and associated stimulation of biological export production: | 100% glacial field |
| 8 | Redfield ratio C:P: | +15% |

**Table 3.** Marine sediment core data and ice core data which were used for determining the parameter changes through a fit of the linear response model. References in parentheses denote the original source of the data in case the data has been taken from a compilation.

| Core name | area | lat. +N, -S | long. -W, +E | depth [m] | record used | reference |
|---|---|---|---|---|---|---|
| Vostok | Antarctica | -78 | -106 | -3488 | at. $CO_2$ | Barnola et al. (1987) |
| M13519 | Eq. Atlantic | 5.333 | -19.183 | n.giv. | $\delta^{13}C_{benthic}$ | Sarnthein et al. (1984) |
| RC13-1110 | Eq. Pacific | 0.100 | -95.650 | 3231 | $\delta^{13}C_{benthic}$ | Mix (1991) |
| EW9209-1JP | N. Atlantic | 5.0 | -43.0 | 4056 | $\delta^{13}C_{benthic}$ | Oliver et al. (2010) (Curry and Oppo (1997)) |
| GEOB1115-3 | S. Atlantic | -3.56 | -12.56 | 2945 | $\delta^{13}C_{benthic}$ | Oliver et al. (2010) (Bickert and Wefer (1996)) |
| GEOB1721-6 | S. Atlantic | -29.17 | 13.08 | 3044 | $\delta^{13}C_{benthic}$ | Oliver et al. (2010) (Bickert and Mackensen (1996)) |
| GEOB1722-3 | S. Atlantic | -29.49 | 11.75 | 3073 | $\delta^{13}C_{benthic}$ | Oliver et al. (2010) (Mollenhauer et al. (2002)) |
| GIK13519-1 | N. Atlantic | 5.67 | -19.85 | 2862 | $\delta^{13}C_{benthic}$ | Oliver et al. (2010) (Sarnthein et al. (1994)) |
| GIK23414-9 | N. Atlantic | 53.54 | -20.29 | 2196 | $\delta^{13}C_{benthic}$ | Oliver et al. (2010) (Jung and Sarnthein (2003c)) |
| GIK23415-9 | N. Atlantic | 53.18 | -19.14 | 2472 | $\delta^{13}C_{benthic}$ | Oliver et al. (2010) (Jung and Sarnthein (2003d)) |
| KNR140-37J | N. Atlantic | 31.41 | -75.26 | 3000 | $\delta^{13}C_{benthic}$ | Oliver et al. (2010) (Keigwin (2004)) |
| ODP1087 | S. Atlantic | -31.46 | 15.31 | 1372 | $\delta^{13}C_{benthic}$ | Oliver et al. (2010) (Pierre et al. (2004) ) |
| GEOB3004-1 | Indian Oc. | 14.61 | 5292. | 1803 | $\delta^{13}C_{benthic}$ | Oliver et al. (2010) (Schmiedl and Mackensen (2006)) |
| MD01-2378 | Indian Oc. | -13.08 | 121.79 | 1783 | $\delta^{13}C_{benthic}$ | Oliver et al. (2010) (Kawamura et al. (2006)) |
| MD97-2121 | S. Pacific | -40.38 | 177.99 | 3014 | $\delta^{13}C_{benthic}$ | Oliver et al. (2010) (Carter et al. (2008)) |
| M13519 | Eq. Atlantic | 5.333 | -19.183 | n.giv. | $\delta^{13}C_{planktonic}$ | Sarnthein et al. (1984) |
| V19-30 | Eq. Pacific | -3.350 | -83.350 | 3091 | $\delta^{13}C_{planktonic}$ | Shackleton and Pisias (1985) |
| GIK17049-6 | N. Atlantic | 55.26 | -26.73 | 3331 | $\delta^{13}C_{planktonic}$ | Oliver et al. (2010) (Jung and Sarnthein (2004)) |
| GIK23415-9 | N. Atlantic | 53.18 | -19.14 | 2472 | $\delta^{13}C_{planktonic}$ | Oliver et al. (2010) (Jung and Sarnthein (2003a)) |
| GIK23418-8 | N. Atlantic | 52.55 | -20.33 | 1491 | $\delta^{13}C_{planktonic}$ | Oliver et al. (2010) (Jung and Sarnthein (2003b)) |
| ODP1089 | S. Atlantic | -40.94 | 9.89 | 4621 | $\delta^{13}C_{planktonic}$ | Oliver et al. (2010) (Hodell et al. (2003b)) |
| RC13-138 | Eq. Pacific | 1.81 | -94.14 | 2655 | $\delta^{13}C_{planktonic}$ | Oliver et al. (2010) (Delphiprojekt, Dürkoop et al. (1997) |
| C4402 | W Pacific | 2.996 | 135.022 | 4402 | BSi | Kawahata et al. (1998) |
| H3571 | N Pacific | 34.904 | 179.703 | 3571 | BSi | Kawahata et al. (2000) |
| RC24-07 | Eq. Atlantic | -1.350 | -11.917 | 3899 | BSi | Verardo and McIntyre (1994) |
| 503B | E. Eq. Pac. | 4.8 | -95.6 | 3672 | BSi | Rea et al. (1986) |

**Table 4.** Marine CaCO$_3$ sediment core data which were used for determining the parameter changes through a fit of the linear response model. References in parentheses denote the original source of the data in case the data has been taken from a compilation.

| Core name | area | lat. +N, -S | long. -W, +E | depth [m] | record used | reference |
|---|---|---|---|---|---|---|
| C4402 | W Pacific | 2.996 | 135.022 | 4402 | CaCO$_3$-wt.% | Kawahata et al. (1998) |
| GC342 | S Pacific | -45.100 | 147.743 | 4002 | CaCO$_3$-wt.% | Moy et al. (2006) |
| H3571 | N Pacific | 34.904 | 179.703 | 3571 | CaCO$_3$-wt.% | Kawahata et al. (2000) |
| MD972106 | S Pacific | -45.148 | 146.285 | 3310 | CaCO$_3$-wt.% | Moy et al. (2006) |
| ODP980 | N Atlantic | 55.485 | -14.702 | 2168 | CaCO$_3$-wt.% | Hyun et al. (2005) |
| RC24-07 | Eq. Atlantic | -1.350 | -11.917 | 3899 | CaCO$_3$-wt.% | Verardo and McIntyre (1994) |
| WEC8803B-GC51 | Eq. Pacific | 1.300 | -133.600 | 4410 | CaCO$_3$-wt.% | LaMontagne et al. (1996) |
| RC11-210 | Eq. Pacific | 1.8 | -140.0 | 4420 | CaCO$_3$-wt.% | Chuey et al. (1987) |
| 503B | E. Eq. Pac. | 4.8 | -95.6 | 3672 | CaCO$_3$-wt.% | Rea et al. (1986) |
| 114-704A | S. Atlantic | -46.879 | 7.421 | 2543 | CaCO$_3$-wt.% | Hoogakker et al. (pers. comm.)(Hodell (1993)) |
| GeoB1032-3 | S. Atlantic | -22.915 | 6.037 | 2502 | CaCO$_3$-wt.% | Hoogakker et al. (pers. comm.)(Bickert (1992); Bickert and Wefer (1996)) |
| GeoB1034-3 | S. Atlantic | -21.735 | 5.422 | 3772 | CaCO$_3$-wt.% | Hoogakker et al. (pers. comm.) (Bickert (1992); Bickert and Wefer (1996)) |
| GeoB1035-4 | S. Atlantic | -21.587 | 5.028 | 4453 | CaCO$_3$-wt.% | Hoogakker et al. (pers. comm.) (Bickert (1992); Bickert and Wefer (1996)) |
| GeoB1041-3 | Eq. Atlantic | -3.475 | -7.60 | 4033 | CaCO$_3$-wt.% | Hoogakker et al. (pers. comm.) (Bickert (1992); Bickert and Wefer (1996); Mackensen and Bickert (1999)) |
| GeoB1105-4 | Eq. Atlantic | -1.665 | -12.428 | 3225 | CaCO$_3$-wt.% | Hoogakker et al. (pers. comm.) (Bickert (1992); Bickert and Wefer (1996); Mackensen and Bickert (1999)) |
| GeoB1112-4 | S. Atlantic | -5.778 | -10.750 | 3125 | CaCO$_3$-wt.% | Hoogakker et al. (pers. comm.) (Bickert (1992); Bickert and Wefer (1996); Mackensen and Bickert (1999)) |
| GeoB1115 | Eq. Atlantic | -3.562 | -12.560 | 2945 | CaCO$_3$-wt.% | Hoogakker et al. (pers. comm.)(Bickert (1992); Bickert and Wefer (1996); Mackensen and Bickert (1999)) |
| GeoB1117-2 | Eq. Atlantic | -3.815 | -14.897 | 3984 | CaCO$_3$-wt.% | Hoogakker et al. (pers. comm.)(Bickert (1992); Bickert and Wefer (1996); Mackensen and Bickert (1999)) |
| GeoB1118 | Eq. Atlantic | -3.560 | -16.428 | 4671 | CaCO$_3$-wt.% | Hoogakker et al. (pers. comm.)(Bickert (1992); Bickert and Wefer (1996); Mackensen and Bickert (1999)) |
| GeoB1211-3 | S. Atlantic | -24.475 | 7.533 | 4084 | CaCO$_3$-wt.% | Hoogakker et al. (pers. comm.)(Bickert (1992); Bickert and Wefer (1996); Mackensen and Bickert (1999)) |
| GeoB1214-1 | S. Atlantic | -24.690 | 7.240 | 3210 | CaCO$_3$-wt.% | Hoogakker et al. (pers. comm.)(Bickert (1992); Bickert and Wefer (1996)) |
| GeoB1505-2 | Eq. Atlantic | 2.267 | -33.015 | 3706 | CaCO$_3$-wt.% | Hoogakker et al. (pers. comm.)(Zabel et al. (1999)) |
| GeoB1710-3 | S. Atlantic | -23.432 | 11.698 | 2987 | CaCO$_3$-wt.% | Hoogakker et al. (pers. comm.)(Schmiedl and Mackensen (1997): Kirst et al. (1999); Kirst (1998)) |
| GeoB1711-4 | S. Atlantic | -23.315 | 12.377 | 1967 | CaCO$_3$-wt.% | Hoogakker et al. (pers. comm.)(Little et al. (1997); Müller (2006)) |
| GeoB3935-2 | N. Atlantic | 12.613 | -59.387 | 1558 | CaCO$_3$-wt.% | Hoogakker et al. (pers. comm.)(Schlünz et al. (2000); Schlunz et al. (2000)) |
| GeoB3938 | N. Atlantic | 12.588 | -58.098 | 2467 | CaCO$_3$-wt.% | Hoogakker et al. (pers. comm.)(Schlünz (1998); Schlunz et al. (2000) |
| GeoB4240-2 | N. Atlantic | 28.888 | -13.225 | 1358 | CaCO$_3$-wt.% | Hoogakker et al. (pers. comm.)(Freudenthal et al. (2002); Freudenthal (1998)) |
| MD95-2039 | N. Atlantic | 40.578 | -10.349 | 3381 | CaCO$_3$-wt.% | Hoogakker et al. (pers. comm.)(Thomson et al. (2000); Thomson et al. (1999); Zahn et al. (1997)) |
| MD96-2080 | S. Atlantic | -36.267 | 19.475 | 2488 | CaCO$_3$-wt.% | Hoogakker et al. (pers. comm.)(Rau et al. (2002)) |
| ODP1088 | S. Atlantic | -41.136 | 13.563 | 2082 | CaCO$_3$-wt.% | Hoogakker et al. (pers. comm.)(Hodell et al. (2003a)) |
| ODP1089 | S. Atlantic | -40.936 | 9.894 | 4620 | CaCO$_3$-wt.% | Hoogakker et al. (pers. comm.)(Hodell et al. (2003a)) |
| PS2082 | S. Atlantic | -43.220 | 11.738 | 4610 | CaCO$_3$-wt.% | Hoogakker et al. (pers. comm.)(Mackensen et al. (1994)) |
| PS2489-2 | S. Atlantic | -42.870 | 8.970 | 3794 | CaCO$_3$-wt.% | Hoogakker et al. (pers. comm.)(Kuhn (1999); Becquey and Gersonde (2002); Becquey and Gersonde (2003)) |
| RC13-228 | S. Atlantic | -22.330 | 11.198 | 3204 | CaCO$_3$-wt.% | Hoogakker et al. (pers. comm.)(Ruddiman and Farrell (1996a); Curry et al. (1988)) |
| TTN057-6 | S. Atlantic | -42.913 | 8.600 | 3751 | CaCO$_3$-wt.% | Hoogakker et al. (pers. comm.)(Hodell et al. (2003a)) |
| V22-108 | S. Atlantic | -43.180 | -3.250 | 4171 | CaCO$_3$-wt.% | Hoogakker et al. (pers. comm.)(Charles et al. (1991)) |
| V25-59 | Eq. Atlantic | 1.370 | -33.480 | 3824 | CaCO$_3$-wt.% | Hoogakker et al. (pers. comm.)(Imbrie (1989)) |
| V30-97 | N. Atlantic | 41.000 | -32.930 | 3371 | CaCO$_3$-wt.% | Hoogakker et al. (pers. comm.)(Ruddiman et al. (1989); McIntyre and Imbrie (2000); McIntyre (1989); Ruddiman and Farrell (1996b)) |
| ELT49018-PC | Indian Oc. | -46.050 | 90.155 | 3282 | CaCO$_3$-wt.% | Hoogakker et al. (pers. comm.)(Howard and Prell (1992)) |
| RC11-120 | Indian Oc. | -43.520 | 79.867 | 3193 | CaCO$_3$-wt.% | Hoogakker et al. (pers. comm.)(Martinson et al. (1987)) |
| CHat01k | S. Pacific | -41.583 | 171.500 | 3556 | CaCO$_3$-wt.% | Hoogakker et al. (pers. comm.)(McCave et al. (2008); Lean and McCave (1998)) |
| CHat3k | S. Pacific | -42.660 | 167.496 | 4802 | CaCO$_3$-wt.% | Hoogakker et al. (pers. comm.)(McCave et al. (2008)) |
| CHat5k | S. Pacific | -40.783 | 171.549 | 4240 | CaCO$_3$-wt.% | Hoogakker et al. (pers. comm.)(McCave et al. (2008)) |
| DSDP594 | S. Pacific | -45.523 | 174.948 | 1204 | CaCO$_3$-wt.% | Hoogakker et al. (pers. comm.)(McCave et al. (2008); ) |
| H3571 | N. Pacific | 34.904 | 179.703 | 3571 | CaCO$_3$-wt.% | Hoogakker et al. (pers. comm.)(Kawahata et al. (2000)) |
| NGC108 | N. Pacific | 36.514 | 158.348 | 3390 | CaCO$_3$-wt.% | Hoogakker et al. (pers. comm.)(Maeda et al. (2002) |
| RC13-115 | Eq. Pacific | -1.652 | -104.800 | 3621 | CaCO$_3$-wt.% | Hoogakker et al. (pers. comm.)(Lyle et al. (2002) |
| TT013-PC72 | Eq. Pacific | 0.114 | -139.402 | 4298 | CaCO$_3$-wt.% | Hoogakker et al. (pers. comm.)(Murray et al. (2000)) |
| V19-27 | Eq. Pacific | -0.467 | -82.070 | 1373 | CaCO$_3$-wt.% | Hoogakker et al. (pers. comm.)(Lyle et al. (2002) |
| V19-28 | Eq. Pacific | -2.367 | -84.650 | 2720 | CaCO$_3$-wt.% | Hoogakker et al. (pers. comm.)(Lyle et al. (2002) |
| V19-30 | Eq. Pacific | -3.383 | -83.520 | 2091 | CaCO$_3$-wt.% | Hoogakker et al. (pers. comm.)(Ninkovich and Shackleton (1975); Shackleton and Pisias (1985)) |
| W8402-14GC | Eq. Pacific | 0.953 | -138.955 | 4287 | CaCO$_3$-wt.% | Hoogakker et al. (pers. comm.)(Lyle et al. (2002) |
| Y69-71 | Eq. Pacific | 0.000 | -86.000 | 2740 | CaCO$_3$-wt.% | Hoogakker et al. (pers. comm.)(Pisias and Mix (1997)) |
| Y71-9-101 | S. Pacific | -6380 | -106.520 | 3175 | CaCO$_3$-wt.% | Hoogakker et al. (pers. comm.)(Lyle et al. (2002); Pisias and Mix (1997)) |

**Table 5.** Maximum parameter changes at last glacial maximum relative to interglacial control run.

| Run specification | SST [°] | land C storage [PgC] | POC dissol. rate [%] | BSi dissol. rate [%] | rain ratio change CaCO3:$C_{org}$ [%] | circulation change on [%] | dust change on [%] | carbon over-consumption change [units C rel. to P] |
|---|---|---|---|---|---|---|---|---|
| Full rank 8 | -6.017 | -416.7 | 5.131 | 3.047 | -34.29 | 48.21 | 7.861 | 28.32 |
| Rank 7 | -7.894 | -441.7 | -1.171 | -11.85 | -7.438 | 25.88 | 9.39 | 10.51 |
| Rank 6 | -4.938 | -438.3 | -0.156 | 2.373 | 1.685 | 45.95 | 12.34 | 10.50 |

**Table 6.** Average goodness of fit expressed as root mean square error for the control run base line and the different optimisations. The values for the optimisations are estimated from the linear fitting procedure.

| Run specification | All records [normalised units] | Atmosp. $CO_2$ [ppm] | $\delta^{13}C$ benthic [‰] | $\delta^{13}C$ planktonic [‰] | BSi [wt-%] | CaCO$_3$ [wt-%] |
|---|---|---|---|---|---|---|
| Control run | 0.4636 | 61.02 | 0.3515 | 0.5515 | 4.454 | 15.10 |
| Full rank 8 | 0.3519 | 21.05 | 0.2443 | 0.4507 | 4.059 | 14.39 |
| Rank 7 | 0.3559 | 38.59 | 0.2452 | 0.4804 | 3.982 | 14.66 |
| Rank 6 | 0.3580 | 53.25 | 0.2446 | 0.4782 | 4.052 | 14.94 |

**Table 7.** Average root mean sqare deviation of the optimised modelled tracer records (as estimated from the simultaneous fit of the sensitivity experiment results to the observations) from the simulated control run tracer record. A zero value would mean that no difference with respect to the control run has occurred. The larger the values, the more differ the optimised tracer curves and the control run tracer curves, i.e. zero change, case.

| Run specification | Atmosp. $CO_2$ [ppm] | $\delta^{13}C$ benthic [‰] | $\delta^{13}C$ planktonic [‰] | BSi [wt-%] | CaCO$_3$ [wt-%] |
|---|---|---|---|---|---|
| Full rank 8 | 46.12 | 0.2471 | 0.3514 | 1.497 | 6.062 |
| Rank 7 | 23.49 | 0.2465 | 0.3831 | 1.090 | 4.752 |
| Rank 6 | 8.887 | 0.2480 | 0.3595 | 1.070 | 3.554 |

**Table A1.** Model parameter values as set for the control or standard run.

| Tunable Model Parameter | parameter symbol | Control Run Value |
|---|---|---|
| specific gas exchange rate for $CO_2$ | $k_{CO2}$ | 0.0603 [mole/($m^2 \cdot$yr$\cdot$ppm)] |
| $O_2$ gas transfer velocity | $k_{O2}$ | 250 [m/yr] |
| threshold value $S_{opal}$ of particle export production ratio C(opal):C(POC) for the onset of $CaCO_3$ production (see eq. 5) | | 0.6 |
| maximum possible production rain ratio C($CaCO_3$):C(POC) (see eq. 12) | $R$ | 0.375 |
| global weathering input of Si | - | $4.5 \cdot 10^{12}$ [mole Si yr$-1$] |
| global weathering input of $CaCO_3$ | - | $27 \cdot 10^{12}$ [mole C yr$^{-1}$] |
| global weathering input of POC | - | $5 \cdot 10^{12}$ [mole C yr$^{-1}$] |
| Si$(OH)_4$ saturation concentration | $C_{sat}$ | 800 [$\mu$mole yr$^{-1}$] |
| degradation rate constant of $CaCO_3$ in water column | $r_{CaCO3}$ | 6.76 [yr$^{-1}$] |
| minimum value for $CaCO_3$ undersaturation in water column | $\Omega_{min}$ | 0.062 |
| degradation rate constant of opal in water column | $r_{opal}$ | 1.23 [yr$^{-1}$] |
| degradation rate constant of POC in water column | $r_{POC}$ | 2.69 [yr$^{-1}$] |
| sinking velocity of particulate matter in water column | $w_M$ | 3 m/day |
| degradation rate constant for dissolved organic carbon in the water column | | 0.05 [yr$^{-1}$] |
| degradation rate constant of $CaCO_3$ in sediment | $r^*_{CaCO3}$ | $6.55 \cdot 10^{-4}$ / C$_{sat,CaCO3}$ [yr$^{-1}$ l mole$^{-1}$] |
| degradation rate constant of opal in sediment | $r^*_{opal}$ | $9.82 \cdot 10^{-4}$ / C$_{sat,opal}$ [yr$^{-1}$ l mole$^{-1}$] |
| degradation rate constant of organic carbon in sediment | $r^*_{orgC}$ | $2.69 \cdot 10^{-4}$ / C$_{sat,opal}$ [yr$^{-1}$ l mole$^{-1}$] |
| diffusion constant for pore waters | $D_w$ | $8 \cdot 10^{-6}$ [cm$^2$/s] |
| coefficient for explicit bioturbation | $D_b$ | 40 [cm$^2$/1000yr] |