# Peer review of "Ocean carbon cycling during the past 130,000 years - a pilot study on inverse paleoclimate record modelling"

_Climate of the Past, 2016_

## Referee Comment (RC1) · D. Archer (Referee) · 13 May 2016

This paper is a clear advance in the question of understanding the glacial / interglacial atmospheric $CO_2$ cycles, as driven (presumably) by the ocean. It's difficult to model this process because the data are impacted by the 3-D circulation of the ocean, but also span a huge dimension of time. So a box model or some of the intermediate complexity models with 2-d ocean are too simple, but a coupled primitive equation climate model would be too slow. The HAMOCC model is an ideal vehicle for exploring this question. Innovations to this work include interpolating the circulation field between glacial and interglacial values, assembly of a suite of paleo data for comparison against, and lots of creative statistical processing for optimizing the model input parameters.

Scaling the flow fields between LGM and today is a clever idea, and worth considering as an interim step as this is. In reality there were certainly fits and starts to the circulation, such as the Heinrich drop-dead mode of the overturning circulation, which will ultimately need to be addressed (by somebody, not necessarily in this paper).

Does the temperature of the deep sea decrease during LGM in the Winguth LGM flow field, the way that Mg/Ca and deep porewater oxygen isotope temperature proxy data suggests that it did? That change in CO2 solubility, and the change in atmosphere / ocean Co2 partitioning, may not be represented in that flow field.

The authors seemed to recoil from the idea that CaCO3 production could have been lower during LGM, because of the expectation from lower CO2 that CO3= would be higher, and thus calcification rates higher. One proposed mechanism to produce a systematic decrease in CaCO3 production was "Silicate leak" from the Southern Ocean, flushing the thermocline with Si which crowded out CaCO3 producers. I'm not advocating that idea, because there's no clear link in sediment traps today between Si / N ratios and the balance between CaCO3 and organic carbon. Another potential CaCO3-decreasing driver is colder temperatures. At any rate, the expected increase in CaCO3 with decreasing CO2 is not really iron-clad either. I understand about the decrease in CaCO3 called for by the inversion not being robust; that is a valid argument. But I don't see that a decrease can be disregarded on first principles.

So the paper could be improved by responding perhaps to these issues and by editing the text for some wordiness and Germanic idiom, but in general the paper represents real progress on a difficult topic, and is clearly worthy of publication.

---

## Referee Comment (RC2) · Anonymous Referee #2 · 25 Jun 2016

Heinze et al. use a coarse resolution ocean biogeochemical model to estimate the effect of changes in SST, terrestrial biosphere release, dissolution rate constant of POC and BSi, CaCO3:POC rain ratio, 3D oceanic velocity field, dust deposition and Redfield C/P ratio on sedimentary d13C, BSi and CaCO3. A linear statistical model is then used to explore the parameter space. The parameters giving the best fit with a range of paleoproxy records are shown.

This is a useful manuscript, which allows the study of a wide range of parameters, but with a "linear response" caveat.

The parameters for the full solution (rank 8) give a large decrease in CaCO3/POC, which the authors suggest is unlikely. They thus decrease the rank to 7, but obtain

too large changes in SST. The final "best" solution is thus the rank 6. But with that solution, there is only little change in atmospheric pCO2. Since the basis of the model is to reproduce a range of paleoproxy records, I am a bit surprised that no model-data comparison are shown for the ranks 8,7 and 6. Shouldn't at least correlation coefficients between model and proxy given for the 3 ranks?

There is limited discussion on previous glacial/interglacial studies, particularly for recent studies, granted the approach used here is quite different.

Figures: Some lines fall out of the y axis range in figures 6 and 7. I understand this is to highlight the fact that CaCO3/POC and SST parameters are going outside the expected range, but aesthetically it is not the best. Also text and lines are sometimes one on top of each other.

Typos: There are a few typos throughout the text and some sentences could be simplified or rewritten for a better flow. Some typos are listed below: p. 6 "EPICA" p15, L16 "on" iof "om" Legend figure 3: "experiment"

---

## Author Comment (AC1) · 28 Jul 2016

**Final author comments for manuscript cp-2016-35**
**Title: Ocean carbon cycling during the past 130,000 years –**
**a pilot study on inverse paleoclimate record modelling**
**by author(s): C. Heinze et al.**

**RESPONSE TO REVIEWER#1 (David Archer):**

We would like to thank the reviewer for the thorough review and the constructive suggestions for improving the manuscript. Below we cite the reviewer's remarks in italics and our direct responses in normal text. The references for the responses to both reviewers are collected together at the end of this document.

*Reviewer#1:*
*This paper is a clear advance in the question of understanding the glacial / interglacial atmospheric*
*$CO_2$ cycles, as driven (presumably) by the ocean. It's difficult to model this process because the data*
*are impacted by the 3-D circulation of the ocean, but also span a huge dimension of time. So a box*
*model or some of the intermediate complexity models with 2-d ocean are too simple, but a coupled*
*primitive equation climate model would be too slow. The HAMOCC model is an ideal vehicle for*
*exploring this question. Innovations to this work include interpolating the circulation field between*
*glacial and interglacial values, assembly of a suite of paleo data for comparison against, and lots of*
*creative statistical processing for optimizing the model input parameters.*

(No response required.)

*Reviewer#1:*
*Scaling the flow fields between LGM and today is a clever idea, and worth considering as an interim*
*step as this is. In reality there were certainly fits and starts to the circulation, such as the Heinrich*
*drop-dead mode of the overturning circulation, which will ultimately need to be addressed (by*
*somebody, not necessarily in this paper).*

Our response:
We plan to add the following text after page 15 line 32 in order to underline that our pragmatic approach can be improved once more realistic ocean velocity fields will be available over the past climatic cycle:
"Ideally, one would need to simulate the glacial ocean circulation in a coupled Earth system model including an ice sheet model over the entire last glacial-interglacial climatic cycle. However, even given such a detailed and realistic velocity would be available, the computing times for carrying out the various sensitivity experiments would be prohibitively large due to the required short time step in such simulations. Nevertheless, in future studies it would be desirable to include also quick alterations of the ocean velocity field, especially changes in ocean overturning. Such short-term climatic changes (time scale of few hundred to thousand years) have been inferred from ice core as well as sediment core analysis known as Dansgaard Oeschger events (Dansgaard et al. (1993), Anklin et al. (1993)) where the coldest events are also marked by large amounts of ice rafted debris in sediment cores (Heinrich events (Bond et al. (1993)). Non-linear ocean-atmosphere dynamics (Barker et al. (2015); Olsen et al. (2005)) would need to be included in respective simulation attempts."
(The new references will be added to the reference list.)

*Reviewer#1:*
*Does the temperature of the deep sea decrease during LGM in the Winguth LGM flow field, the way*
*that Mg/Ca and deep pore water oxygen isotope temperature proxy data suggests that it did? That*
*change in $CO_2$ solubility, and the change in atmosphere / ocean $CO_2$ partitioning, may not be*
*represented in that flow field.*

Our response:
After the new text as cited above, we plan to insert the following passage for clarification and critical appraisal:
"Also the representation of sea water temperature changes can be improved. The LGM sea surface temperatures have been accounted for through the respective glacial forcing field underlying the simulation (CLIMAP Project Members (1976); CLIMAP Project Members (1981)). The simulated deep water temperature drop below 1500 m was around 1.2∘C on the average (Winguth et al. (1999)) as compared to the pre-industrial/interglacial simulation, with some areas where the temperature difference was up to -2∘C, especially in the North Atlantic deep water. Reconstructions of bottom water temperatures through oxygen isotope pore water analysis revealed a temperature decrease of around 2∘C at the Carnegie Ridge (Pacific) and the Ceara Rise (Atlantic) (Cutler et al. (2003)) and close to deep water productions sites cooling of deep waters in North Atlantic, South Pacific, and Southern Ocean by about 4-5∘C, 2.5∘C, and 1.5∘C (Adkins et al. (2002)). Consistent with this, bottom water interglacial-glacial temperature changes have been inferred from Mg/Ca paleo-thermometry (Dwyer et al. (1995), Skinner et al. (2003), Roberts et al. (2016)). The modelled sea water temperatures may thus be somewhat higher than the observed ones, especially for the Southern Ocean. It should be note that the circulation change experiment with the biogeochemical model was carried out with preindustrial temperatures (for the biogeochemistry only) in order to separate the temperature and circulation effects properly (and to avoid linear parameter dependencies in the inverse approach)."
(The new references will be added to the reference list.)

*Reviewer#1:*
*The authors seemed to recoil from the idea that $CaCO_3$ production could have been lower during LGM, because of the expectation from lower $CO_2$ that $CO_3{}^{2-}$ would be higher, and thus calcification rates higher. One proposed mechanism to produce a systematic decrease in $CaCO_3$ production was "Silicate leak" from the Southern Ocean, flushing the thermocline with Si which crowded out $CaCO_3$ producers. I'm not advocating that idea, because there's no clear link in sediment traps today between Si / N ratios and the balance between $CaCO_3$ and organic carbon. Another potential $CaCO_3$-decreasing driver is colder temperatures. At any rate, the expected increase in $CaCO_3$ with decreasing $CO_2$ is not really iron-clad either. I understand about the decrease in $CaCO_3$ called for by the inversion not being robust; that is a valid argument. But I don't see that a decrease can be disregarded on first principles.*

Our response:
Yes, we agree that the rain ratio hypothesis cannot be discarded. To make this clearer, we plan to modify line 27 on page 14 and insert the following text (after: "The rain ratio change shows a dramatic decrease in pelagic $CaCO_3$ production"):

"Such a change may not be fully out of scope (see, e.g., the discussion in Broecker and Peng (1986) and Berger and Keir (1984)). Archer and Maier-Reimer (1994) argued that enhanced $CaCO_3$ dissolution on the sea floor through organic carbon degradation in combination with a rain ratio reduction would provide an efficient way for reducing atmospheric $pCO_2$. The rain ratio change itself could be provided by an increased surface ocean concentration of silicic acid by which diatoms would dominate over $CaCO_3$ shell material production. Such a change in silicic acid concentration could be induced by enhanced iron fluxes to the Southern Ocean by dust, thinner opal frustules after the iron stress has been diminished and subsequent export of "unused" silicic acid from the Southern Ocean to the rest of the world ocean ("silicic acid leakage hypothesis", Matsumoto et al. (2002); Griffiths et al. (2013)). Further it has been argued that low seawater temperatures lead to lower water column remineralisation rates for organic carbon and changes in the ecosystem community structure that would imply a rain ratio reduction (Matsumoto et al. (2007)). On the other hand, in case of a strong coupling between deep POC fluxes to $CaCO_3$ fluxes (where $CaCO_3$ works as ballast for downward organic carbon transport; see Klaas and Archer (2002) and Armstrong et al. (2002)), rain ratio shifts at the ocean surface would only have a minor impact on atmospheric $pCO_2$ (Ridgwell

(2003)). Further, one would expect rather an increase in $CaCO_3$ production at low ambient $pCO_2$ and high $CaCO_3$ saturation (Zondervan et al. (2001); Riebesell et al. (2000)). Therefore,… "
(The new references will be added to the reference list.)

*Reviewer#1:*
*So the paper could be improved by responding perhaps to these issues and by editing the text for some wordiness and Germanic idiom, but in general the paper represents real progress on a difficult topic, and is clearly worthy of publication.*

Responses to the scientific issues are given above. The language issues will be removed.

(The references for the citations above are listed in the response to reviewer#2 together with the references for that reviewer.)

---

## Author Comment (AC2) · 28 Jul 2016

**Final author comments for manuscript cp-2016-35**
**Title: Ocean carbon cycling during the past 130,000 years – a pilot study on inverse paleoclimate record modelling**
**by author(s): C. Heinze et al.**

**RESPONSE TO REVIEWER#2:**

We also would like to thank reviewer#2 for the expert review and the constructive comments for improving the manuscript. Below we cite the reviewer's remarks in italics and our direct responses in normal text.

*Reviewer#2:*
*Heinze et al. use a coarse resolution ocean biogeochemical model to estimate the effect of changes in SST, terrestrial biosphere release, dissolution rate constant of POC and BSi, CaCO₃:POC rain ratio, 3D oceanic velocity field, dust deposition and Redfield C/P ratio on sedimentary d13C, BSi and CaCO₃. A linear statistical model is then used to explore the parameter space. The parameters giving the best fit with a range of paleoproxy records are shown.*

(No response required.)

*Reviewer#2:*
*This is a useful manuscript, which allows the study of a wide range of parameters, but with a "linear response" caveat.*
*The parameters for the full solution (rank 8) give a large decrease in CaCO₃/POC, which the authors suggest is unlikely. They thus decrease the rank to 7, but obtain too large changes in SST. The final "best" solution is thus the rank 6. But with that solution, there is only little change in atmospheric pCO₂. Since the basis of the model is to reproduce a range of paleoproxy records, I am a bit surprised that no model data comparison is shown for the ranks 8, 7 and 6. Shouldn't at least correlation coefficients between model and proxy given for the 3 ranks?*

We have computed correlation coefficients between observations and predicted values from the linear response model for each paleoclimatic record employed in the fitting procedure for the three ranks considered (and also smaller ranks). We then summarised the findings in a diagram for full rank, rank 7, and rank 6, structured into tracer types and ranks. We plan to insert this figure as Figure 11. We plan to include the following text passage after page 17 line 24:
"Correlation coefficients between observations and predicted tracer values from the linear response model are given in Figure 11. While for a series of single records the correlation is good, for other single records the correlation is poor or even anti-correlations resulted. This is especially the case for the planktonic $\delta^{13}C_{planktonic}$ records. This deficiency can be in part explained through the coarse model resolution. If the regional extent of upwelling zones is different between model and reality also the respective surface tracers for paleo-productivity as recorded in the sediment cores will show respective differences."
(A respective figure caption will be added.)

*Reviewer#2:*
*There is limited discussion on previous glacial/interglacial studies, particularly for recent studies, granted the approach used here is quite different.*

We plan to add the following text in the discussion section before the start of the conclusion section, (i.e., after the text as suggested in the response to the previous referee comment):
"In previous studies on the glacial-interglacial changes in the ocean carbon cycle, often hypotheses involving one specific mechanism were presented or the multitude of potential mechanisms was reviewed. Studies were the simultaneous contributions from several processes to glacial carbon dynamics have been discussed are relatively scarce. Brovkin et al. (2007) employed an Earth system

model of intermediate complexity including oceanic and terrestrial biogeochemical modules to test the impact of simultaneous changes on the atmospheric $CO_2$ concentration. Their results are fairly consistent with those of this study. According to Brovkin et al. (2007), largest contributions to the $CO_2$ drawdown came from circulation and SST changes as well as from a strengthening of the biological pump through improved nutrient utilisation, while the land outgassing amounted to an atmospheric $pCO_2$ increase by 15 ppm. Rain ratio changes contributed to about 15 ppm, a process also cited as a less certain mechanism by Brovkin et al. (2007). In addition they report secondary changes of atmospheric $pCO_2$ due to weathering, sea level change, and changes in sedimentation (shallow water vs. deep water).

Recent studies focused again on the mineral dust hypothesis involving increased iron supply especially to the Southern Ocean (originally revived by Martin et al. (1994) and Berger and Wefer (1991)) and a respective regional strengthening of biological production and carbon export. Increased LGM aerosol iron flux to the Southern Ocean could be corroborated by Conway et al. (2015). With our approach, in a separate experiment (not shown in this study where we only consider dust for solution of sedimentary material) we also had tested increased ocean productivity and related surface nutrient drawdown due to changing dust deposition. The inverse approach did not favour this process. This is fairly consistent with the modelling study by Lambert et al. (2015) arriving at a direct effect of the iron induced biological pump strengthening of less than 10 ppm and delayed effect due to carbon burial and carbonate compensation by about 10 ppm.

Inspired by the suggestion of temperature-dependent export production of Laws et al. (2000), Matsumoto et al. (2007) quantify in a further single mechanism study the effect of temperature dependent remineralisation on the atmospheric $CO_2$ using an ocean biogeochemical model. According to their findings, an LGM atmospheric $CO_2$ decline by 30 ppm would be possible through this process. In our study, the parameter change of the POC remineralisation rate with temperature forcing did not result in a similarly likely drawdown when tested in the inverse approach against evidence from the sedimentary record. Rather our work suggests that simple temperature and $pCO_2$ dependent changes of ocean physics as well as biogeochemistry do not straightforwardly translate into atmospheric $CO_2$ changes and respective sedimentary imprints, but that the problem is more complicated.

The Southern Ocean has been established to be one of the key regions for regulating glacial-interglacial carbon dynamics. Apart from processes involving dissolved iron and nutrients, especially the physical dynamical processes - and hence stratification, deep water production, upwelling, water mass formation, and lateral advection - have been considered in conjunction with the physical/chemical and biological carbon pumps. Special attention has been placed on a northward shift of the westerlies wind forcing at the LGM as compared to today (Toggweiler et al. (2006); Watson and Garabato (2006); Watson et al. (2015)). The general idea is that a northward shift of Southern Ocean upwelling leads to reduced $CO_2$ outgassing and enhanced carbon export to the deep waters resulting in a deep ocean accumulation of organic matter from the surface and hence a vertical "fractionation" of carbon as well as nutrients as described already by Boyle (1988a) and Boyle (1988b)). This general view is corroborated also by recent proxy data findings (for the Southern Ocean by Gottschalk et al., 2014; also for the North Atlantic by Hoogakker et al., 2015). Refined Southern Ocean dynamics could also improve the results of our studies, but we have been limited to the flow fields available. In general, Southern Ocean flow field and tracer simulations show traditionally a large intermodel spread (Broecker et al. (1998); Roy et al. (2011); Orr (2002)). One of the reasons is the complex interplay of sea-ice as well as wind forcing and also the subgrid-scale parameterisation of convection events which occur on narrow regional scales in reality (Gordon (1978)). Still, simulating the Southern Ocean flow field and mixing remains a key challenge even for the modern ocean (Farneti et al. (2015): Downes et al. (2015); Mignot et al. (2013); Abernathey et al. (2016))."

*Reviewer#2:*
*Figures: Some lines fall out of the y axis range in figures 6 and 7. I understand this is to highlight the fact that CaCO₃/POC and SST parameters are going outside the expected range, but aesthetically it is not the best. Also text and lines are sometimes one on top of each other.*

Technically improved figures will be provided.

*Reviewer#2:*
*Typos: There are a few typos throughout the text and some sentences could be simplified or rewritten for a better flow. Some typos are listed below: p. 6 "EPICA" p15, L16 "on" iof "om" Legend figure 3: "experiment"*

The language issues will be addressed together with the respective remarks by Reviewer#1.

[revised manuscript text omitted]